# ARMAP: Scaling autonomous agents via Automatic Reward Modeling And Planning

**Zhenfang Chen**[*]
MIT-IBM Watson AI Lab

**Delin Chen**[*]
UMass Amherst

**Rui Sun**[*]
University of California, Los Angeles

**Wenjun Liu**[*]
UMass Amherst

**Chuang Gan**
UMass Amherst and MIT-IBM Watson AI Lab

## Abstract

Large language models (LLMs) have demonstrated remarkable capabilities across a range of text-generation tasks. However, LLMs still struggle with problems requiring multi-step decision-making and environmental feedback, such as online shopping, scientific reasoning, and mathematical problem-solving. Unlike pure text data, collecting large-scale decision-making data is challenging. Moreover, many powerful LLMs are only accessible through APIs, which hinders their fine-tuning for agent tasks due to cost and complexity. To address LLM agents' limitations, we propose a framework that can automatically learn a reward model from the environment without human annotations. This model can be used to evaluate the action trajectories of LLM agents and provide heuristics for task planning. Specifically, our approach involves employing one LLM-based agent to navigate an environment randomly, generating diverse action trajectories. Subsequently, a separate LLM is leveraged to assign a task intent and synthesize a negative response alongside the correct response for each trajectory. These triplets (task intent, positive response, and negative response) are then utilized as training data to optimize a reward model capable of scoring action trajectories. This reward model can be integrated with LLM-based agents and various planning algorithms to enhance task-solving performance. The effectiveness and generalizability of our framework are demonstrated through evaluations conducted on different agent benchmarks. In conclusion, our proposed framework represents a significant advancement in enhancing LLM agents' decision-making capabilities. By automating the learning of reward models, we overcome the challenges of data scarcity and API limitations, potentially revolutionizing the application of LLMs in complex and interactive environments. This research paves the way for more sophisticated AI agents capable of tackling a wide range of real-world problems requiring multi-step decision-making.[1]

## 1 Introduction

Developing AI agents capable of perceiving environments, understanding instructions, and acting to accomplish a wide range of tasks in interactive settings (Brooks, 1986) have many real-world applications, including virtual human assistants (Reed et al., 2022; Casheekar et al., 2024), business process management (Kirchdorfer et al., 2024), and robotic process automation (Hong et al., 2023b; Rana et al., 2023; Palo et al., 2023).

The recent advent of large generative models has revolutionized numerous applications, such as question answering (Rajpurkar et al., 2016), text summarization (Hermann et al., 2015), and multimodal understanding (Goyal et al., 2017; Yu et al., 2016; Chen et al., 2024). However, while these models excel in comprehension and generation tasks, their performance in decision-making scenarios—such as online shopping and scientific reasoning falls relative short of human capabilities. This disparity likely stems from the nature of the training data. Large generative models are typically

---

[*]Equal contribution.
[1]Project page: https://armap-agent.github.io

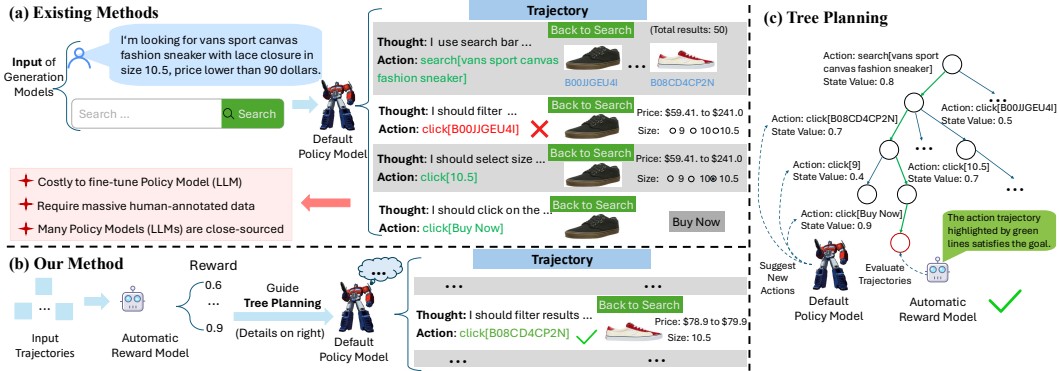

Figure 1: In Fig. 1 (a), we show that it is difficult for LLM agents to generate multi-step plans in an interactive environment to achieve the instruction goal. However, it is relatively easy for an LLM to learn a reward model that can evaluate whether the trajectories meet the task instructions, as shown in Fig. 1 (b). In Fig. 1 (c), we show that a learned reward model can be used to guide the default policy models to improve action planning.

pre-trained on readily available image and text corpora from the internet. In contrast, trajectory data for agent tasks, which require multi-step interaction with the environment, is more challenging to collect and does not naturally occur on the internet. Furthermore, current state-of-the-art commercial LLMs, such as GPT-4V (OpenAI et al., 2024) and Gemini (Reid et al., 2024), only provide limited APIs for general users. This restriction renders it either infeasible or cost-prohibitive to fine-tune these models for specific agent tasks, further impeding progress in this field.

Previous studies have explored the development of autonomous agents for decision-making tasks using large language models (LLMs). Early research (Yao et al., 2023a; Zheng et al., 2024; Deng et al., 2024) utilized instruction prompts with few-shot examples to direct LLMs in handling various agent tasks. These methods do not require task-specific fine-tuning but have shown limited performance on benchmarks requiring interaction with environments and precise action prediction. A different research direction involves collecting human preference data (Hong et al., 2023a) or distilling trajectory data from advanced commercial LLM APIs (Zeng et al., 2023; Deng et al., 2024) and fine-tuning smaller open-source LLMs to create new policy models for agent tasks. However, this distillation process relies on advanced pre-trained agent models for trajectory data extraction, which are often unavailable, expensive, or subject to commercial restrictions. For instance, data from models such as GPT-4 or Gemini cannot be used for commercial purposes.

A fundamental premise of our approach is that, in most agent applications, evaluation is easier than generation (Karp, 1975; Naor, 1996). As illustrated in Fig. 1 (a), *generating* a correct multi-step solution to navigate to the target page is challenging since it needs to predict multiple actions and interact with the environment. However, it is relatively simple to *evaluate* whether the output action trajectories and environment states meet the provided intent to find a *"vans sport canvas fashion sneaker"*. Building on this premise, we suggest that developing a reward model is more feasible than creating a policy model for agent tasks. With an effective reward model, it becomes possible to guide LLMs in planning tasks both effectively and efficiently. For instance, as depicted in Fig. 1 (c), by integrating the reward model with an LLM-based agent and the Monte Carlo Tree Search (MCTS) algorithm (Silver et al., 2017; Coulom, 2006), we can simulate and evaluate the future states of agent tasks, thereby making better decisions for subsequent actions. This approach is analogous to mental simulation (Hegarty, 2004; Lake et al., 2017) in cognitive science, where humans envision the outcomes of potential actions to make better decisions in problem-solving.

While reward models can assist LLM agents in planning, developing these reward models presents significant challenges. Some prior studies have utilized powerful commercial LLM APIs as evaluators for tasks (Kwon et al., 2023a). Although these approaches have demonstrated effectiveness in certain applications, they rely on state-of-the-art LLM models for evaluation, which are often expensive and difficult to scale. In this paper, we introduce an automated method to learn multi-modal reward models without relying on state-of-the-art LLMs for guidance. Furthermore, previous work has not considered integrating the learned reward models with various planning algorithms for problem-solving.

The process of learning the reward model involves three steps. Initially, we utilize an LLM-based agent (e.g., Dubey et al. (2024)) to navigate in the environments, aiming to achieve a randomly proposed intent while collecting extensive action trajectory demonstrations. Subsequently, the LLM examines the collected trajectories and proposes a refined intent that the sampled trajectories actually accomplish. Additionally, we prompt the LLM to generate negative trajectories that fail to achieve the intended task. Finally, based on the synthetic data (intents, positive trajectories, and negative trajectories) collected, we train a customized reward model using widely adopted vision-language models such as VILA (Lin et al., 2023) to evaluate whether the user's intent has been fulfilled by the action trajectories. With this automatic reward model, we enhance the performance of LLM-based agents in conjunction with various planning algorithms such as best of n, reflexion, and MCTS.

In summary, we introduce a novel framework ARMAP (autonomous Agents from automatic Reward Modeling And Planning) for LLM-based agents incorporating an automatic reward model that evaluates task completion, analogous to mental simulation in human cognition. This framework offers several advantages: (1) Effectiveness: It enhances the performance of various LLM agents across different tasks. (2) Flexibility: It eliminates the need for fine-tuning the LLMs themselves and allows for optimization of custom reward targets during inference, enabling more controllable generation. (3) Practicality: The training of the automatic reward model does not rely on labor-intensive labeling or state-of-the-art commercial LLMs, making it more feasible and widely applicable.

## 2 RELATED WORK

**LLMs for Agent tasks.**   Our research is related to deploying large language models (LLMs) as agents for decision-making tasks in interactive environments (Liu et al., 2023b; Zhou et al., 2023; Shridhar et al., 2020; Toyama et al., 2021). Earlier works, such as (Yao et al., 2023a), fine-tuned models like BERT (Devlin et al., 2019) for decision-making in simplified environments, such as online shopping or mobile phone manipulation. With the advent of large language models (Brown et al., 2020; OpenAI et al., 2024), it became feasible to perform decision-making tasks through zero-shot or few-shot in-context learning. To better assess the capabilities of LLMs as agents, several models have been developed (Deng et al., 2024; Xiong et al., 2024; Hong et al., 2023a; Yan et al., 2023). Most approaches (Zheng et al., 2024; Deng et al., 2024) provide the agent with observation and action history, and the language model predicts the next action via in-context learning. Additionally, some methods (Zhang et al., 2023a; Li et al., 2023; Song et al., 2024) attempt to distill trajectories from state-of-the-art language models to train more effective policy models. In contrast, our paper introduces a novel framework that automatically learns a reward model from LLM agent navigation, using it to guide the agents in making more effective plans.

**LLM Planning.**   Our paper is also related to planning with large language models. Early researchers (Brown et al., 2020) often prompted large language models to directly perform agent tasks. Later, Yao et al. (2022) proposed ReAct, which combined LLMs for action prediction with chain-of-thought prompting (Wei et al., 2022). Several other works (Yao et al., 2023b; Hao et al., 2023; Zhao et al., 2023; Qiao et al., 2024) have focused on enhancing multi-step reasoning capabilities by integrating LLMs with tree search methods. Our model differs from these previous studies in several significant ways. First, rather than solely focusing on text generation tasks, our pipeline addresses multi-step action planning tasks in interactive environments, where we must consider not only historical input but also multimodal feedback from the environment. Additionally, our pipeline involves automatic learning of the reward model from the environment without relying on human-annotated data, whereas previous works rely on prompting-based frameworks that require large commercial LLMs like GPT-4 (OpenAI et al., 2024) to learn action prediction. Furthermore, ARMAP supports a variety of planning algorithms beyond tree search.

**Learning from AI Feedback.** In contrast to prior work on LLM planning, our approach also draws on recent advances in learning from AI feedback (Bai et al., 2022; Lee et al., 2023; Yuan et al., 2024; Sharma et al., 2024; Pan et al., 2024; Koh et al., 2024b). These studies initially prompt state-of-the-art large language models to generate text responses that adhere to predefined principles and then potentially fine-tune the LLMs with reinforcement learning. Like previous studies, we also prompt large language models to generate synthetic data. However, unlike them, we focus not on fine-tuning a better generative model but on developing a classification model that evaluates how well action trajectories fulfill the intended instructions. This approach is simpler, requires no reliance on state-of-the-art LLMs, and is more efficient. We also demonstrate that our learned reward model can integrate with various LLMs and planning algorithms, consistently improving their performance.

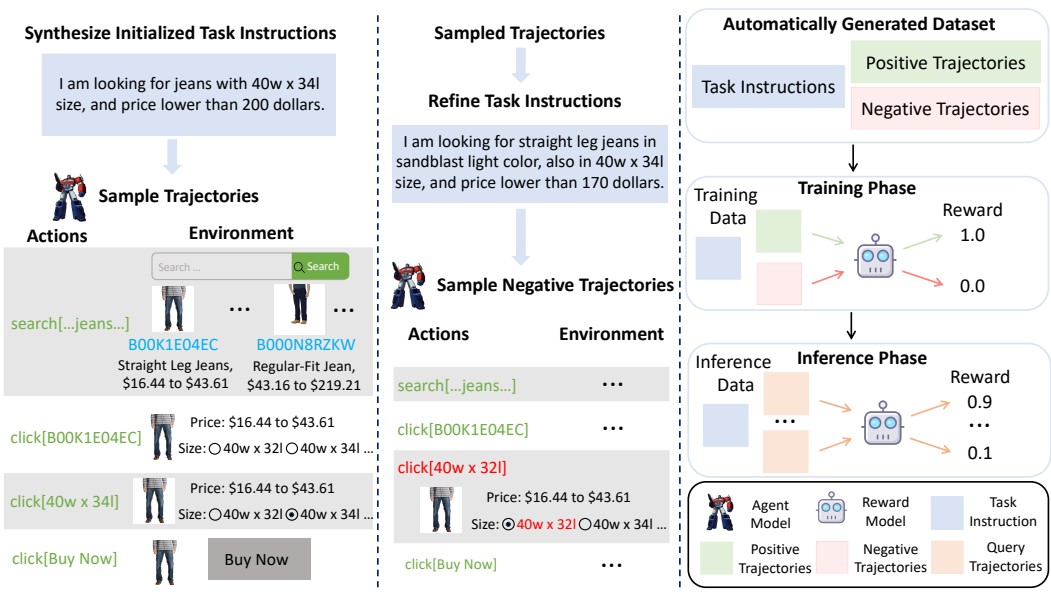

Figure 2: The pipeline of our ARMAP framework. We first generate an initial task instruction using LLMs with in-context learning and sample trajectories aligned with the initial language instructions in the environment. Next, we use the LLM to summarize the sampled trajectories and generate refined task instructions that better match these trajectories. We then modify specific actions within the trajectories to perform new actions in the environment, collecting negative trajectories in the process. Using the refined task instructions, along with both positive and negative trajectories, we train a lightweight reward model to distinguish between matching and non-matching trajectories. The learned reward model can then collaborate with various LLM agents to improve task planning.

**Inference-Time Scaling.** Snell et al. (2024) validates the efficacy of inference-time scaling for language models. Based on inference-time scaling, various methods have been proposed, such as random sampling (Wang et al., 2022b) and tree-search methods (Hao et al., 2023; Zhang et al., 2024a; Guan et al., 2025; Zhang et al., 2023b). Concurrently, several works have also leveraged inference-time scaling to improve the performance of agentic tasks. Koh et al. (2024b) adopts a training-free approach, employing MCTS to enhance policy model performance during inference and prompting the LLM to return the reward. Gu et al. (2024) introduces a novel speculative reasoning approach to bypass irreversible actions by leveraging LLMs or VLMs. It also employs tree search to improve performance and prompts an LLM to output rewards. Yu et al. (2024) proposes Reflective-MCTS to perform tree search and fine-tune the GPT model, leading to improvements in Koh et al. (2024a). Putta et al. (2024) also utilizes MCTS to enhance performance on web-based tasks such as Yao et al. (2023a) and real-world booking environments. Lin et al. (2025) utilizes the stepwise reward to give effective intermediate guidance across different agentic tasks. Our work differs from previous efforts in two key aspects: (1) Broader Application Domain. Unlike prior studies that primarily focus on tasks from a single domain, our method demonstrates strong generalizability across web agents, mathematical reasoning, and scientific discovery domains, further proving its effectiveness. (2) Flexible and Effective Reward Modeling. Instead of simply prompting an LLM as a reward model, we finetune a small scale VLM (Lin et al., 2023) to evaluate input trajectories.

## 3 MODEL

In this section, we provide a detailed introduction to our framework, autonomous Agents from automatic Reward Modeling And Planning (ARMAP). The framework includes automated reward data generation in section 3.2, reward model design in section 3.3, and planning algorithms in section 3.4.

### 3.1 BACKGROUND

The planning tasks for LLM agents can be typically formulated as a Partially Observable Markov Decision Process (POMDP): $(\mathcal{X}, \mathcal{S}, \mathcal{A}, \mathcal{O}, \mathcal{T})$, where:

- $\mathcal{X}$ is the set of text instructions;

- $\mathcal{S}$ is the set of environment states;

- $\mathcal{A}$ is the set of available actions at each state;

- $\mathcal{O}$ represents the observations available to the agents, including text descriptions and visual information about the environment in our setting;

- $\mathcal{T} : \mathcal{S} \times \mathcal{A} \to \mathcal{S}$ is the transition function of states after taking actions, which is given by the environment in our settings.

Given a task instruction $x \in \mathcal{X}$ and the initial environment state $s_0 \in \mathcal{S}$, planning tasks require the LLM agents to propose a sequence of actions $\{a_n\}_{n=1}^N$ that aim to complete the given task, where $a_n \in \mathcal{A}$ represents the action taken at time step $n$, and $N$ is the total number of actions executed in a trajectory. Following the $n$-th action, the environment transitions to state $s_n$, and the agent receives a new observation $o_n$. Based on the accumulated state and action histories, the task evaluator determines whether the task is completed.

An important component of our framework is the learned reward model $\mathcal{R}$, which estimates whether a trajectory $h$ has successfully addressed the task:

$$r = \mathcal{R}(x, h), \tag{1}$$

where $h = \{\{a_n\}_{n=1}^N, \{o_n\}_{n=0}^N\}$, $\{a_n\}_{n=1}^N$ are the actions taken in the trajectory, $\{o_n\}_{n=0}^N$ are the corresponding environment observations, and $r$ is the predicted reward from the reward model. By integrating this reward model with LLM agents, we can enhance their performance across various environments using different planning algorithms.

## 3.2 AUTOMATIC REWARD DATA GENERATION.

To train a reward model capable of estimating the reward value of history trajectories, we first need to collect a set of training language instructions $\{x_m\}_{m=1}^M$, where $M$ represents the number of instruction goals. Each instruction corresponds to a set of positive trajectories $\{h_m^+\}_{m=1}^M$ that match the instruction goals and a set of negative trajectories $\{h_m^-\}_{m=1}^M$ that fail to meet the task requirements. This process typically involves human annotators and is time-consuming and labor-intensive (Christiano et al., 2017; Rafailov et al., 2024). As shown in Fig. 8 of the Appendix. we automate data collection by using Large Language Model (LLM) agents to navigate environments and summarize the navigation goals without human labels.

**Instruction Synthesis.** The first step in data generation is to propose a task instruction for a given observation. We achieve this using the in-context learning capabilities of LLMs. The prompt for instruction generation is shown in Fig. 9 of the Appendix. Specifically, we provide some few-shot examples in context along with the observation of an environment state to an LLM, asking it to summarize the observation and propose instruction goals. In this way, we collect a set of synthesized language instructions $\{x_m^{raw}\}_{m=1}^M$, where $M$ represents the total number of synthesized instructions.

**Trajectory Collection.** Given the synthesized instructions $x_m^{raw}$ and the environment, an LLM-based agent is instructed to take actions and navigate the environment to generate diverse trajectories $\{x_m^{raw}, h_m\}_{m=0}^M$ aimed at accomplishing the task instructions. Here, $h_m$ represents the $m$-th history trajectory, which consists of $N$ actions $\{a_n\}_{n=1}^N$ and $N+1$ environment observations $\{o_n\}_{n=0}^N$. Due to the limited capabilities of current LLMs, the generated trajectories $h_m$ may not always align well with the synthesized task instructions $x_m$. To address this, we ask the LLM to summarize the completed trajectory $h_m$ and propose a refined goal $x_m^r$. This process results in a set of synthesized demonstrations $\{x_m^r, h_m\}_{m=0}^{M_r}$, where $M_r$ is the number of refined task instructions.

**Pairwise Data Construction.** To train a reward model capable of distinguishing between good and poor trajectories, we also need trajectories that do not satisfy the task instructions. To create these, we sample additional trajectories that differ from $\{x_m^r, h_m\}$ and do not meet the task requirements by modifying actions in $h_m$ and generating corresponding negative trajectories $\{h_m^-\}$. For clarity, we refer to the refined successful trajectories as $\{x_m, h_m^+\}$ and the unsuccessful ones as $\{x_m, h_m^-\}$. These paired data will be used to train the reward model described in Section 3.3, allowing it to estimate the reward value of any given trajectory in the environment.

## 3.3 Reward Model Design.

**Reward Model Architectures.** Theoretically, we can adopt any vision-language model (Liu et al., 2023a; Li et al., 2024b;a; Lin et al., 2023) that can take a sequence of visual and text inputs as the backbone for the proposed reward model. In our implementation, we use the recent VILA model (Lin et al., 2023) as the backbone for reward modeling since it has carefully maintained open-source code, shows strong performance on standard vision-language benchmarks like (Fu et al., 2023; Goyal et al., 2017; Hudson & Manning, 2019), and support multiple image input.

The goal of the reward model is to predict a reward score to estimate whether the given trajectory $(x_m, h_m)$ has satisfied the task instruction or not, which is different from the original goal of VILA models that generate a series of text tokens to respond to the task query. To handle this problem, we additionally add a fully-connected layer for the model, which linearly maps the hidden state of the last layer into a scalar value.

**Optimazation Target.** Given the pairwise data that is automatically synthesized from the environments in Section 3.2, we optimize the reward model by distinguishing the good trajectories $(x_m, h_m^+)$ from bad ones $(x_m, h_m^-)$. Following standard works of reinforcement learning from human feedback (Bradley & Terry, 1952; Sun et al., 2023b;a), we treat the optimization problem of the reward model as a binary classification problem and adopt a cross-entropy loss. Formally, we have

$$\mathcal{L}(\theta) = -\mathbf{E}_{(x_m, h_m^+, h_m^-)}[\log \sigma(\mathcal{R}_\theta(x_m, h_m^+) - \mathcal{R}_\theta(x_m, h_m^-))], \quad (2)$$

where $\sigma$ is the sigmoid function and $\theta$ are the learnable parameters in the reward model $\mathcal{R}$. By optimizing this target, the reward model is trained to give higher value scores to the trajectories that are closer to the goal described in the task instruction.

## 3.4 Planning with Large Vision-Langauge Reward Model.

After getting the reward model to estimate how well a sampled trajectory match the given task instruction, we are able to combine it with different planning algorithms to improve LLM agents' performance. Here, we summarize the typical algorithms we can adopt in this paper.

**Best of N.** This is a simple algorithm that we can adopt the learned reward model to improve the LLM agents' performances. We first prompt the LLM agent to generate $n$ different trajectories independently and choose the one with the highest predicted reward score as the prediction for evaluation. Note that this simple method is previously used in natural language generation (Zhang et al., 2024b) and we adopt it in the context of agent tasks to study the effectiveness of the reward model for agent tasks.

**Reflexion.** Reflexion (Shinn et al., 2024) is a planning framework that enables large language models (LLMs) to learn from trial-and-error without additional fine-tuning. Instead of updating model weights, Reflexion agents use verbal feedback derived from task outcomes. This feedback is converted into reflective summaries and stored in an episodic memory buffer, which informs future decisions. Reflexion supports various feedback types and improves performance across decision-making, coding, and reasoning tasks by providing linguistic reinforcement that mimics human self-reflection and learning.

**MCTS.** We also consider tree search-based planning algorithms like Monte Carlo Tree Search (MCTS) (Coulom, 2006; Silver et al., 2017) to find the optimal policy. There is a tree structure constructed by the algorithm, where each node represents a state and each edge signifies an action. Beginning at the initial state of the root node, the algorithm navigates the state space to identify action and state trajectories with high rewards, as predicted by our learned reward model.

The algorithm tracks 1) the frequency of visits to each node and 2) a value function that records the maximum predicted reward obtained from taking action $a$ in state $s$. MCTS would visit and expand nodes with either higher values (as they lead to high predicted reward trajectory) or with smaller visit numbers (as they are under-explored). We provide more details in the implementation details and the appendix section.

## 4 Experiments

In this section, we conduct a series of experiments to demonstrate the effectiveness of the proposed framework for agent tasks. First, we evaluate the framework's performance on standard agent benchmarks (Yao et al., 2023a; Wang et al., 2022a; Yao et al., 2023b), detailed in Section 4.2. Next,

we show how customizing the reward target during inference allows us to generate more tailored action plans, as described in Section 4.3. Finally, we conduct ablation studies in Section 4.4. Before delving into the experimental results, we provide an overview of our experimental setup.

## 4.1 EXPERIMENTAL SETUP

**Environments.** We evaluate the ARMAP framework in three different environments:

- **Webshop** is a well-known environment for online shopping (Yao et al., 2023a), where the agent must search for and select products on the website to obtain a final result. Following the setup of AgentBench (Liu et al., 2023b) for LLM evaluation, we test the model on the validation split, using the default matching reward as the evaluation metric.

- **ScienceWorld** (Wang et al., 2022a) is an interactive benchmark designed for embodied science experiments. It places agents in a simulated text-based environment where they must perform elementary science experiments by navigating the environment, manipulating objects, and observing outcomes. The aim is to assess whether AI models can apply scientific knowledge, rather than merely retrieve or assemble information. We evaluate the framework on both seen and unseen splits.

- **Game of 24** is a mathematical game where the agent is given four numbers and must use arithmetic operations (addition, subtraction, multiplication, and division) to make the number 24. For instance, given the input '3, 5, 7, 11,' one possible solution is '$(7-3)*(11-5) = 24$'. Following Yao et al. (2023b), we selected 100 challenging puzzles, specifically those indexed from 901 to 1,000, and the performance metric is the success rate across these puzzles. As shown in Fig. 7 of the Appendix, we use the chain-of-thought prompting technique, prompting the LLM agents to output intermediate steps followed by the final answer. Each step of the solution is considered an action.

**LLM Setup.** Our framework requires LLM models to act as agents, generating synthetic task instructions from the environment along with few-shot examples in the prompt context. We also deploy agents to perform these synthetic tasks in the environment, collecting diverse trajectories for further analysis. In this paper, we primarily use the Llama3-70b-instruct model (Dubey et al., 2024) to synthesize training data for the automatic reward models, as it is open-source, easy to deploy locally, and delivers robust performance. We avoid state-of-the-art commercial models like GPT-4 or Gemini due to their high costs and the complexity of reproducing results caused by frequent model updates, making them less suitable for our research objectives.

To evaluate the performance of various LLM agents, we serve a representative set of LLM APIs locally, balancing model diversity with affordable serving costs. We identify the LLMs by their model family and size. Specifically, these are Llama70B, Llama8B, Mistral7B, and Phi3.8B. We note that these open-source model families are frequently updated, and we provide the current model links in the Appendix A.3. All models can be easily set up using the vLLM library (Kwon et al., 2023b) and a single H100 GPU.

**Baselines.** We implement our ARMAP framework using different planning algorithms, including Reflexion, Best-of-N, and MCTS, which we denote as **ARMAP-R**, **ARMAP-B**, and **ARMAP-M**, respectively. We limit the maximum number of trajectories our ARMAP can explore to 10 in the ScienceWorld and Webshop environments to systematically evaluate the pipeline's effectiveness across different LLM agent backbones. We also compare the model with two baselines that do not use reward model guidance: **Sampling** and **Greedy**. For the Game of 24 environment, we follow the setup of a previous study (Yao et al., 2023b) and set the maximum number of explored trajectories to 100. For **Sampling**, we set the model temperature to 1 and sample action trajectories using chain-of-thought prompting (Wei et al., 2023). For **Greedy**, we set the temperature to 0, generating the action sequence with the highest probability. Further implementation details are provided in the Appendix. We will release all the code, model, and data for easy reproduction upon acceptance.

## 4.2 EFFECTIVENESS FOR REWARD PLANNING.

In this section, we investigate the effectiveness of the framework across different language models (Dubey et al., 2024; Jiang et al., 2023; Abdin et al., 2024) and various planning algorithms. The results are shown in Table 1. Based on the table, we have the following observations. First, our proposed pipeline is effective, as it consistently outperforms the **Sampling** and **Greedy** baselines

| Backbone | Algorithms | Webshop | ScienceWorld seen | unseen | Game24 | Average |
|----------|-----------|---------|------|--------|--------|---------|
| Llama70B | Sampling | 52.0 | 53.9 | 50.6 | 9.6 | 38.0 |
|          | Greedy   | 50.4 | 57.2 | 55.1 | 6.0 | 37.5 |
|          | ARMAP-R  | 56.5 | **59.0** | 56.7 | 16.0 | 43.5 |
|          | ARMAP-B  | 62.0 | 57.3 | **57.0** | 19.0 | 46.1 |
|          | ARMAP-M  | **66.8** | 58.2 | 55.9 | **24.0** | **49.3** |
| Llama8B  | Sampling | 56.4 | 24.5 | 20.6 | 2.0 | 27.0 |
|          | Greedy   | 57.7 | 29.9 | 23.8 | 2.0 | 28.9 |
|          | ARMAP-R  | 58.3 | 31.2 | 28.0 | 6.0 | 31.3 |
|          | ARMAP-B  | 59.3 | **35.7** | **28.1** | **11.0** | **34.1** |
|          | ARMAP-M  | **60.2** | 32.5 | 24.9 | 9.0 | 32.6 |
| Mistral7B | Sampling | 17.7 | 18.4 | 17.1 | 1.0 | 12.2 |
|          | Greedy   | 37.2 | 21.1 | 19.6 | 1.0 | 19.5 |
|          | ARMAP-R  | 54.1 | 21.7 | 19.7 | 2.0 | 25.6 |
|          | ARMAP-B  | 54.4 | 24.5 | 21.2 | 2.0 | 26.4 |
|          | ARMAP-M  | **58.2** | **30.0** | **23.4** | **4.0** | **29.6** |
| Phi3.8B  | Sampling | 34.7 | 10.0 | 7.6 | 2.0 | 15.2 |
|          | Greedy   | 42.4 | 9.5 | 6.5 | 2.1 | 17.5 |
|          | ARMAP-R  | 53.3 | 9.6 | 7.2 | 4.0 | 21.9 |
|          | ARMAP-B  | 52.1 | 20.0 | 17.0 | 9.0 | 26.5 |
|          | ARMAP-M  | **53.7** | **28.3** | **24.3** | **10.0** | **30.0** |

Table 1: Effectiveness of the proposed method on different benchmarks. Our ARMAP framework consistently outperforms the baselines across different language models.

across different planning algorithms. Additionally, we observe that the average improvement is more significant on weaker models, such as Phi (Abdin et al., 2024) and Mistral-7B (Jiang et al., 2023), compared to stronger models like Llama3-1-70B (Dubey et al., 2024). We believe this is because weaker models explore more low-reward trajectories, providing greater opportunities for the reward model to improve performance.

Among the three planning algorithms, MCTS performs the best on average, likely due to its superior mechanisms for identifying higher-reward trajectories and searching less-explored trajectories. We also notice that Reflexion performs the worst on weaker models like Mistral7B and Phi3.8B. We suspect this is because Reflexion was designed for ChatGPT-family-based agents and requires the LLM agent to possess strong capabilities for learning from trial and error. Finally, we present qualitative results of different methods in Fig. 3, where it is clear that our ARMAP generates better trajectories than the baselines, aided by the guidance of automatic reward models. In Appendix A.5, we analyze several failure cases, offer more detailed insights into the limitations of the current approach, and suggest potential improvements in reward modeling.

### 4.3 CONTROLLABLE GENERATION.

Another benefit of our ARMAP pipeline is that we can customize our reward targets during inference, allowing us to generate more controllable action sequences, rather than solely maximizing the predicted rewards. Agent fine-tuning methods (Li et al., 2023; Zeng et al., 2023) find it challenging to achieve this goal since agent behaviors are typically fixed during inference. We conducted experiments in the Webshop environment to evaluate the impact of customizable reward targets. In addition to the original objective of maximizing the predicted reward $\mathcal{R}(x, h)$, we defined two additional optimization targets. First, we aimed to minimize the number of actions in the trajectory history, defining the reward target as $\mathcal{R}(x, h) - \text{NumberOfAction}(h)$. Second, we sought to minimize the price of the target product, with a customized target of $\mathcal{R}(x, h) - \text{PriceOfProduct}(h)$. Table 2 presents the results. By applying a length penalty on the reward target for ARMAP-M, we reduced the average action length from 4.5 to 4 and the average product price from 97.9 to 69.0, while maintaining comparable performance on the default matching reward. Similar performance was observed for ARMAP-B. Additionally, we provide a qualitative example in Fig. 4. From this example, we can see that our customized reward target successfully guided the LLM agent to purchase products with fewer action steps while still finding the target product.

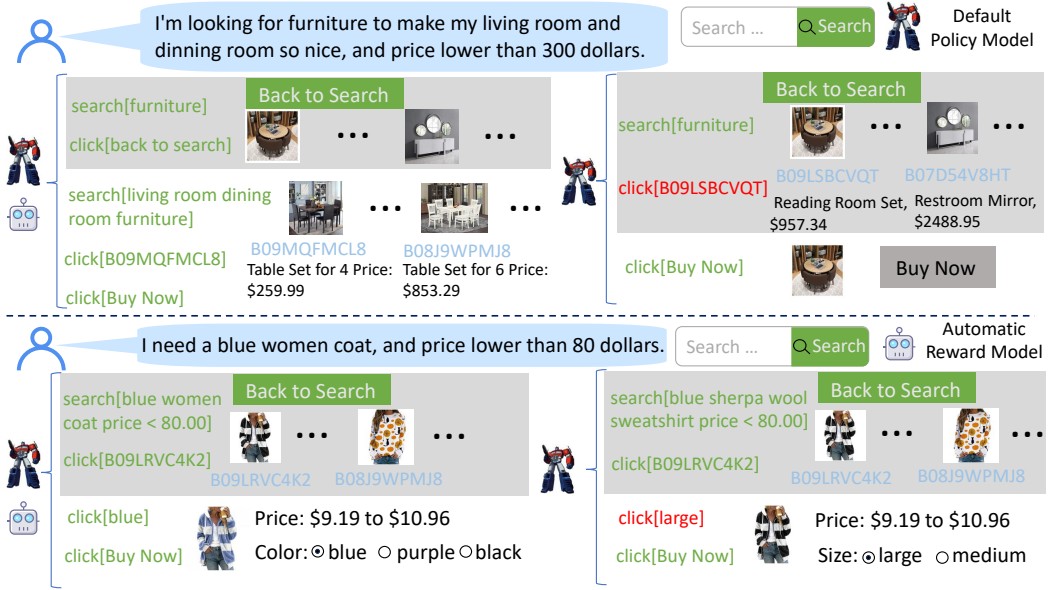

Figure 3: Two qualitative results of the Webshop task. The figure shows two examples utilizing the advantages of our ARMAP framework and we are able to correct errors made by existing methods. In the top example, when the search results do not meet the requirements, our ARMAP method leverages the advantage of the tree structure to backtrack and search again, thereby retrieving the appropriate target item. In contrast, existing methods fail to backtrack when the target item is not found. In the bottom example, by using the ARMAP to evaluate different states in the environment, our method is able to select the color that offers a higher reward and better meets the requirements when choosing between size and color, rather than mistakenly selecting the wrong size. These two examples sufficiently demonstrate the advantages of our method compared to traditional approaches.

| Algorithms | Action↓ | Price ↓ | Reward ↑ |
|---|---|---|---|
| Greedy | 4.6 | 102.4 | 50.4 |
| ARMAP-B | 4.7 | 102.2 | 62.0 |
| ARMAP-M | 4.5 | 97.9 | 66.8 |
| ARMAP-B+Length-Penalty | 3.9 | 98.8 | 60.3 |
| ARMAP-M+Length-penalty | 4.0 | 102.1 | 65.5 |
| ARMAP-B+Price-penalty | 5.0 | 65.5 | 57.5 |
| ARMAP-M+Price-penalty | 4.3 | 69.0 | 62.4 |

Table 2: Controllable Trajectory Generation. We show that we can generate controllable trajectories like shorter action lengths and lower prices by customizing reward targets. We use Llama70B as the default API for action prediction.

## 4.4 ABLATION STUDIES.

We conduct ablation studies to investigate the effectiveness of the framework. Specifically, we aim to answer the following questions: **Q1.** Can we train a policy model with fully supervised learning to handle multi-step tasks from the synthesized trajectory data? **Q2.** Can a large, general language model be used as the reward model to perform guidance without automatic reward learning?

We conducted experiments using the ScienceWorld benchmark, and the results are shown in Table 3. When comparing our pipeline to the SFT model trained using our reward backbone VILA3B, we observed that although the policy model trained through fully supervised learning performed reasonably well (18.6), it still lagged behind the performance of our planning framework (28.3). This suggests that learning a policy model is more challenging than learning a reward model, highlighting the effectiveness of our proposed ARMAP pipeline (answering **Q1**).

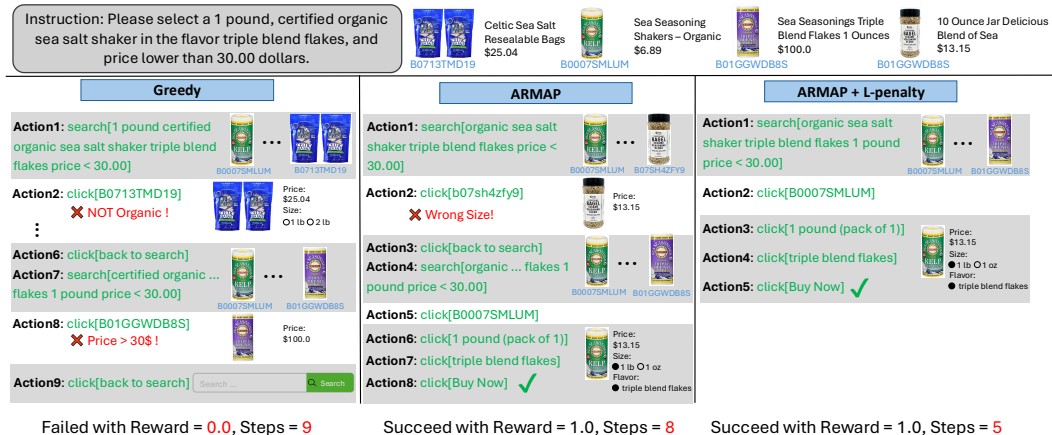

Figure 4: A typical example of customized reward target for shorter trajectory generation. On the left, we show the default greedy decoding generates a long trajectory without finding the target product. In the middle, we show our default reward can guide the LLM agent to generate a correct but long trajectory. On the right, we show our framework with a customized reward target for shorter trajectories, which finds a correct and short trajectory for the target product.

| Models | Model Base | ScienceWorld ( seen ) |
|--------|-----------|----------------------|
| Greedy | Phi3.8B | 9.6 |
| SFT-Policy | VILA3B | 18.6 |
| ARMAP-B w/o R | Llama70B and Phi3.8B | 16.0 |
| ARMAP-M w/o R | | 26.5 |
| ARMAP-B | VILA3B and Phi3.8B | 20.0 |
| ARMAP-M | | 28.3 |

Table 3: Ablation study of the proposed framework. Our ARMAP framework is more effective than directly finding a policy model and using the general LLM for reward generation.

Next, we replaced our smaller 3B reward model with a much larger language model, Llama3-1-70B, and used few-shot prompting to predict the reward of the extracted trajectories. We found that this larger model also improved performance compared to the default greedy model, demonstrating the effectiveness of our planning framework. However, it still performed worse than our pipeline using automatic reward learning, despite the Llama3-1-70B being about 20 times larger, further showcasing the efficiency and effectiveness of our approach (answering **Q2**).

We provide additional ablation experiments in the Appendix A.2, including the data quality from various LLMs, reward modeling target and computational efficiency.

## 5 CONCLUSION

We propose a framework, ARMAP, for large language model (LLM) agents to manage tasks that require multi-step decision-making and environmental feedback, such as online shopping or scientific reasoning. This framework allows LLM-based agents to enhance task planning by autonomously learning a reward model from the environment, without the need for human labeling. The method utilizes pre-trained LLM agents to generate diverse action trajectories within an environment, which are then evaluated by a separate LLM based on the task's intent. These evaluations help train a reward model that strengthens the agents' decision-making capabilities. The framework enhances the performance of LLM agents in addressing complex tasks and mitigates issues related to data scarcity and API limitations. Its effectiveness is demonstrated across various benchmarks, representing a significant advancement in the development of AI agents for real-world, multi-step problem-solving.

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

# A   APPENDIX

In this section, we provide supplementary material for the main paper.

## A.1   EXPERIMENTS ON ALFWORLD AND AGENTCLINIC.

We extend our experiment on ALFWorld (Shridhar et al., 2021), a classic environment for House-Holding, where the agent must accomplish tasks in physical house-holding environments, like "Put a pan on the dining table". Following the setup of AgentBench (Liu et al., 2023b) for LLM evaluation, we test the model on the dev and std split, using the default success rate as the evaluation metric. Specifically, we used LLaMa-3.1-70B to generate around 1600 pairs of positive and negative samples with our data generation pipeline. Then we train a reward model with these synthesized data. We evaluate our ARMAP framework on ALFWorld using various planning algorithms, including Reflexion and Best-of-N, which we refer to as ARMAP-R and ARMAP-B, respectively. Additionally, we compare our approach with two baseline methods that do not incorporate reward model guidance: Sampling and Greedy. The results are shown below. As shown in Table 4, our model still performs well in this challenging environment, which contains diverse scenes and long-horizon planning tasks.

| Models | ALFWorld-std | ALFWorld-dev |
|---|---|---|
| Sampling | 0.13 | 0.14 |
| Greedy | 0.18 | 0.30 |
| ARMAP-R | 0.22 | 0.35 |
| ARMAP-B | 0.30 | 0.45 |

Table 4: Experimental Results on ALFWorld.

We also extended our experiments to ClinicalAgent (Schmidgall et al., 2024), an environment designed for medical decision-making tasks. ClinicalAgent evaluates models on their ability to interpret clinical scenarios and make accurate, high-stakes decisions. Results of ClinicalAgent are provided in Table 5, further supporting the versatility of ARMAP in domains requiring precise reasoning.

| Models | AgentClinic-MedQA |
|---|---|
| Sampling | 11.89 |
| Greedy | 14.02 |
| ARMAP-B | 44.33 |

Table 5: Experiments Results on AgentClinic.

## A.2   ABLATION STUDY.

**Dependence on Quality of Synthetic Data from Various LLMs**. We choose ScienceWorld and conduct experiments to study the effectiveness of different reward models. As shown in Table 6, the left column represents the results of using LLaMA-8B greedy directly and the Best of N results of LLaMA-8B with the reward model trained by the data generated from LLaMA-70B, LLaMA-8B, Mistral-7B, and Phi-3.8B, respectively. Greedy is our baseline result and it can be observed that using the reward model leads to better experimental outcomes. Among all the results, LLaMA-70B achieves the best performance. Compared to the other three models, LLaMA-70B has the largest scale and is naturally the most capable model. LLaMA-8B and Mistral-7B have a similar number of parameters, and in the ScienceWorld task, Mistral-7B performs better than LLaMA-8B. Phi-3.8B is the smallest of these models, yet it still achieved very good results. Notably, compared to the larger-scale LLaMA-8B and Mistral-7B, Phi-3.8B still scored better. These results indicate that our method exhibits good robustness when faced with LLMs of different scales and capabilities. Even with the smallest model, our method can still achieve good results. From these experimental

| Models | SciWorld-seen | SciWorld-unseen |
|--------|---------------|-----------------|
| Greedy | 29.9 | 23.8 |
| Llama70B | 35.7 | 28.1 |
| Llama8B | 32.2 | 24.7 |
| Mistral7B | 33.7 | 26.5 |
| Phi3.8B | 34.7 | 26.9 |

Table 6: Experiments of training data generated from various LLMs.

outcomes, it is clear that our method does not overly rely on the capabilities of language models. In other words, our method is highly efficient and robust.

**Reward Modeling Target.** To further investigate the optimization target of the reward model, we conduct experiments to compare the performance of pairwise comparison and binary classification as learning methods for the reward model. Specifically, in the classification setting: each input pair is treated as a positive and a negative example. The model is trained to predict a score of 1 for positive examples and 0 for negative examples. The comparative results are shown in Table 7. Across all settings, pairwise comparison consistently outperforms binary classification. This confirms that pairwise comparison captures nuanced preferences more effectively than binary classification, leading to better reward modeling and overall task performance.

| Backbone | Algorithms | Classification | | Comparative | |
|----------|-----------|-------|--------|------|--------|
| | | Seen | Unseen | Seen | Unseen |
| LLaMA-70B | ARMAP-R | 57.0 | 55.4 | 59.0 | 56.7 |
| | ARMAP-B | 47.2 | 43.3 | 57.3 | 57.0 |
| LLaMA-8B | ARMAP-R | 29.0 | 24.2 | 31.2 | 28.0 |
| | ARMAP-B | 27.5 | 22.2 | 35.7 | 28.1 |
| Mistral-7B | ARMAP-R | 17.8 | 18.2 | 21.7 | 19.7 |
| | ARMAP-B | 19.1 | 17.3 | 24.5 | 21.1 |
| Phi-3.8B | ARMAP-R | 8.6 | 4.8 | 9.6 | 7.2 |
| | ARMAP-B | 17.7 | 13.7 | 20.0 | 17.0 |

Table 7: Comparison of the Classification target and Comparison target on ScienceWorld.

**Computational Efficiency Analysis.** We further study the data demands of the reward modelings. We show the performance of using different amounts of training data. In Table 8 and Table 9, we selected ScienceWorld and used ARMAP-B as the experimental setting. In the leftmost column, we listed the different LLMs used in our study. In the first row, we introduced VILA-3B, VILA-13B, and LLaVA-13B, to compare the impact of different sizes and types of reward models on the final outcomes. In the last two columns, we trained the reward models using 1/5 and 1/25 of the original training dataset size, respectively, to assess how varying amounts of training data affect our method. (1) As seen, the effectiveness of our method continues to improve with increasing reward model sizes. However, in the experiments with LLaMA-8B and Phi-3.8B, despite using more potent reward models, there was no improvement in results. We believe that in the processes of planning and reasoning, the capability of the policy model still plays a dominant role. If the policy model is more robust, and concurrently, if we enhance the capability of the reward model, we can continuously achieve better results. (2) We also observe that the performance of LLaVA-13B is not as good as VILA-13B. We attribute this to VILA being an improved version of LLaVA, and it utilizes an interleaved image-text dataset in its training, which better aids the model in perceiving, understanding, and handling multimodal information. Hence, VILA outperforms LLaVA. (3) From the Table 8 and Table 9, it is evident that regardless of whether the data is seen or unseen, increasing the model size improves the final experimental results. If we use the results of the VILA-3B model as a benchmark and compare it with the two settings, 1/5 data and 1/25 data, it is clear that increasing the training data enhances the outcomes. Conversely, even when using extremely limited data amounts like 1/5

or 1/25 of the original dataset, we can still achieve a capable model, and the performance does not dramatically decrease.

These results demonstrate that our method can still yield good results in a low-resource environment. In other words, our approach does not rely on large volumes of data and the strong capability of large models; it is succinct and efficient, capable of performing well in extremely low-resource settings.

| Backbone | VILA-3B | VILA-13B | LLaVA-13B | 1/5 Data | 1/25 Data |
|----------|---------|----------|-----------|----------|-----------|
| LLaMA-70B | 57.3 | 61.2 | 44.3 | 52.1 | 50.6 |
| LLaMA-8B | 35.7 | 34.3 | 26.0 | 31.4 | 29.3 |
| Mistral-7B | 24.5 | 26.0 | 19.5 | 22.6 | 21.7 |
| Phi-3.8B | 20.0 | 19.5 | 16.7 | 17.9 | 13.9 |

Table 8: Comparison of reward model selection and data demands on ScienceWorld seen set.

| Backbone | VILA-3B | VILA-13B | LLaVA-13B | 1/5 Data | 1/25 Data |
|----------|---------|----------|-----------|----------|-----------|
| LLaMA-70B | 57.0 | 60.7 | 48.2 | 50.0 | 47.7 |
| LLaMA-8B | 28.1 | 27.5 | 22.2 | 26.8 | 24.2 |
| Mistral-7B | 21.1 | 22.9 | 19.2 | 21.6 | 19.7 |
| Phi-3.8B | 17.0 | 15.3 | 13.7 | 14.2 | 11.7 |

Table 9: Comparison of reward model selection and data demands on ScienceWorld unseen set.

**Ablation on Visual Input**. We also train a new reward model without visual information. As shown in Table 10, we can see that, in different settings, the reward model with visual information performs better than the model without visual information, which shows the importance of visual context in the Webshop task.

| Backbone | Algorithms | w/o visual | w/ visual |
|----------|-----------|------------|-----------|
| LLaMA-70B | ARMAP-R | 56.1 | 56.5 |
| | ARMAP-B | 61.6 | 62.0 |
| Mistral-7B | ARMAP-R | 53.6 | 54.1 |
| | ARMAP-B | 51.3 | 54.4 |

Table 10: Ablation of the visual input.

**Overhead in Data Synthesis.** We calculate the tokens we have used for task instruction generation and trajectory exploration. We summarize these overheads in Table 11. To provide a more intuitive comparison, we first calculated the average tokens per sample for these different tasks. We found that although Game of 24 overall consumes the most tokens, the average number of tokens spent per Game of 24 sample is relatively the least. In contrast, Webshop has the fewest total samples but the highest average number of tokens spent per sample. ScienceWorld falls in between these two. The reason Webshop has a higher average number of tokens compared to Game of 24 is that the environment required for Webshop is more complex, involving more diverse elements and possibilities.

| Tasks | Samples | Tokens | Tokens per Sample |
|-------|---------|--------|-------------------|
| ScienceWorld | 4064 | 2541255 | 625 |
| Webshop | 2436 | 6645746 | 2728 |
| Game of 24 | 37885 | 12846182 | 339 |

Table 11: Tokens of data generation in three different tasks.

**Proprietary Models as Training Data Generators and Policy Models.** In the main content, we mainly consider using open-source models to act as training data generators and policy models. In

order to investigate the upper bounds of our proposed method, we also conduct some experiments by using powerful proprietary models. However, to serve as the training data generator, closed-source models have several drawbacks, including high costs, limited commercial access, and lack of reproducibility. In contrast, our approach achieves strong results without relying on closed-source models. Given the expense associated with API-based models like GPT-4o for generating training datasets, we have opted not to pursue this method for now.

For API-based proprietary models serving as policy models, the high cost of GPT-4o and API access rate limitations prompted us to focus our experiments primarily on ALFWorld. Specifically, we used GPT-4o-2024-08-06 to sample five trajectories each on ALFWorld's Dev and Std sets, then conducted experiments using our automatic reward model. As shown in Table 12, our reward model is able to help the powerful GPT-4o gain better performance, demonstrating the effectiveness of our framework.

| GPT-4o | Std | Dev |
|---------|------|------|
| Sampling | 0.74 | 0.88 |
| Greedy | 0.82 | 0.90 |
| ARMAP-B | 0.84 | 0.95 |

Table 12: Experiments of using the proprietary model on ALFWorld

### A.3 Implementation Details.

**Large Pretrain Model Setup.** We serve a diverse set of open-source LLM APIs to evaluate the effectiveness of the proposed pipeline. We list all the open-source models and their weights on huggingface in table 13. All these models can be easily setup and reproduced with the VLLM libarary (Kwon et al., 2023b). We prove the effectiveness of our ARMAP framework across different LLM APIs.

| Acronym | Model description and weight on huggingface websites |
|---------|------------------------------------------------------|
| Llama70B | https://huggingface.co/hugging-quants/Meta-Llama-3.1-70B-Instruct-AWQ-INT4 |
| Llama8B | https://huggingface.co/meta-llama/Meta-Llama-3.1-8B-Instruct |
| Mistral7B | https://huggingface.co/mistralai/Mistral-7B-Instruct-v0.3 |
| Phi3.8B | https://huggingface.co/microsoft/Phi-3.5-mini-instruct |
| VILA3B | https://huggingface.co/Efficient-Large-Model/VILA1.5-3b |

Table 13: Agent models, the reward model, and their associated description on huggingface websites.

**Environment Setup.** We build our environments based on the environment setup of the previous works (Liu et al., 2023b; Song et al., 2024; Yao et al., 2023b; Shridhar et al., 2021; Schmidgall et al., 2024). For Webshop and ALFWorld environment, we start these docker environments from AgentBench (Liu et al., 2023b) and implement different planning algorithms, Reflexion, Best-of-N and MCTS on it. Similarly, we build our ScienceWorld, Game of 24 and AgentClinic environments from Song et al. (2024), Yao et al. (2023b) and Schmidgall et al. (2024), respectively.

**Planning Algorithm Details.** We compare the performance of different planning algorithms by limiting their maximum explored trajectory number. We set the maximum number to be 10 on Webshop and ScieneWorld in consideration of effectiveness and efficiency. We set the maximum number to be 100 on Game of 24 following the setup of Yao et al. (2023b). In Webshop, ScienceWorld, ALFWorld and AgentClinic benchmarks, we only consider the top 10 available actions suggested by the LLM agent at each state to reduce search space. We also set a trajectory's maximal action number length to 10 for simplicity.

For Reflexion, we set the maximum trial number to be 10 for all tasks. For different tasks and models, we set the threshold of Reflexion separately. During the iteration process, if the reward of the current

trail's trajectory exceeds the threshold, the iteration will stop, and the current trail will be taken as the result. If the maximum number of trials is reached, the last trial will be taken as the result in Webshop and Game of 24, while the first trial will be taken as the result in ScienceWorld.

**Data Generation.** In total, we generate 2,436, 4,064 and 37,885 pairs of data for Webshop, ScienceWorld and Game of 24, respectively. Sampled synthesized data sample can be seen in Fig. 5, Fig. 6 and Fig. 7. We provide the sampled prompt we use for data generation from Fig. 8 to Fig. 11. In Fig. 8, we show an example how we prompt the LLM to generate language instruction for ScienceWorld. In Fig. 9, we show how we refine the language instruction to refine the instruction goal. In Fig. 10 and Fig. 11, we show the prompt how the LLM agent synthesizes positive and negative trajectories, respectively.

---

Training Data Example for Webshop

**Task Instruction:** I need 2 long-lasting bluetooth radio speaker units for a soundbar with 4ohm impedance and 6w power, and price lower than 50.00 dollars.

**Positive Example:**
**Thought:** ...
**Action:** ...
**Current Screenshot:** ...
**Observation:** ...
**Thought:** The option B09STMXYR5 matches the required characteristics. I should select it.
**Action:** click[B09STMXYR5]
**Current Screenshot:**

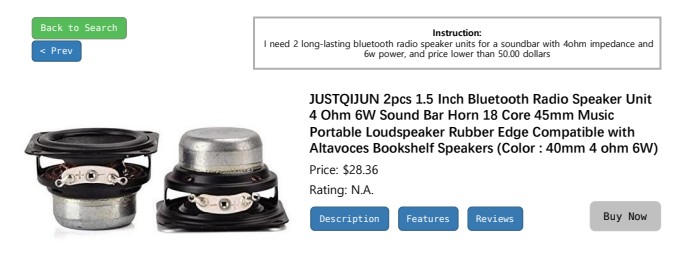

**Observation:** Instruction: [SEP] i need 2 long-lasting bluetooth radio speaker units for a soundbar with 4ohm impedance and 6w power, and price lower than 50.00 dollars [SEP] Back to Search [SEP] < Prev [SEP] JUSTQIJUN 2pcs 1.5 Inch Bluetooth Radio Speaker Unit 4 Ohm 6W Sound Bar Horn 18 Core 45mm Music Portable Loudspeaker Rubber Edge Compatible with Altavoces Bookshelf Speakers (Color : 40mm 4 ohm 6W) [SEP] Price: $28.36 [SEP] Rating: N.A. [SEP] Description [SEP] Features [SEP] Reviews [SEP] Buy Now
**Thought:** I should click on the 'Buy Now' button to proceed with the purchase.
**Action:** click[buy now]

**Negative Example:**
**Thought:** ...
**Action:** ...
**Current Screenshot:** ...
**Observation:** ...
**Thought:** I think I should click on the product I need, which is B09SWKXBY5, since it matches the requirements of 4ohm impedance and 6w power and the price is lower than 50.00 dollars.
**Action:** click[B09SWKXBY5]
**Current Screenshot:**

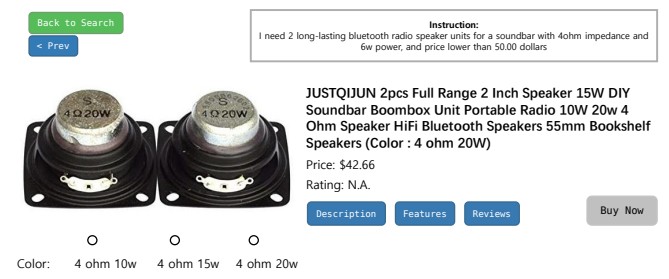

**Observation:** Instruction: [SEP] i need 2 long-lasting bluetooth radio speaker units for a soundbar with 4ohm impedance and 6w power, and price lower than 50.00 dollars [SEP] Back to Search [SEP] < Prev [SEP] color [SEP] 4 ohm 10w [SEP] 4 ohm 15w [SEP] 4 ohm 20w [SEP] JUSTQIJUN 2pcs Full Range 2 Inch Speaker 15W DIY Soundbar Boombox

---

Unit Portable Radio 10W 20w 4 Ohm Speaker HiFi Bluetooth Speakers 55mm Bookshelf Speakers (Color : 4 ohm 20W) [SEP] Price: $42.66 [SEP] Rating: N.A. [SEP] Description [SEP] Features [SEP] Reviews [SEP] Buy Now
**Thought:** I think I should click on the 'Buy Now' button to purchase the product.
**Action:** click[buy now]

Figure 5: Training Data Example for Webshop.

### Training Data Example for ScienceWorld

**Task Instruction:** You are a helpful assistant to do some scientific experiments in an environment. In the environment, there are several rooms: kitchen, foundry, workshop, bathroom, outside, living room, bedroom, greenhouse, art studio, hallway You should explore the environment and find the items you need to complete the experiment. You can teleport to any room in one step. All containers in the environment have already been opened, you can directly get items from the containers.
The available actions are:
open OBJ: open a container
close OBJ: close a container
...
wait1: task no action for a step
Task Description: Your task is to find a non-living thing in the environment. Focus on the jug in the greenhouse. When you are done, move the jug to the green box in the kitchen.

**Positive Example:**
**Action:** look around
**Observation:** This room is called the hallway. In it, you see:
the agent, a substance called air, a picture
You also see:
A door to the art studio (that is open), A door to the bedroom (that is open), A door to the greenhouse (that is open), A door to the kitchen (that is open), A door to the living room (that is open), A door to the workshop (that is open)
**Action:** teleport to the greenhouse
**Observation:** You teleport to the greenhouse.
**Action:** look around
**Observation:** This room is called the greenhouse. In it, you see:
the agent, a substance called air, a adult bee, a adult bee, a adult bee, a adult bee, a bee hive. The bee hive door is open. In the bee hive is: a adult bee. a flower pot 2 (containing a peach tree in the adult stage, soil), a flower pot 4 (containing a substance called water), a flower pot 5 (containing a substance called water), a flower pot 6 (containing a banana tree in the adult stage, soil), a flower pot 7 (containing a apple tree in the adult stage, soil), a jug (containing nothing), a shovel, a sink, which is turned off. In the sink is: nothing.
You also see:
A door to the hallway (that is open), A door to the outside (that is open)
**Action:** focus on the jug
**Observation:** You focus on the jug.

**Negative Example:**
**Action:** look around
**Observation:** This room is called the hallway. In it, you see:
the agent, a substance called air, a picture
You also see:
A door to the art studio (that is open), A door to the bedroom (that is open), A door to the greenhouse (that is open), A door to the kitchen (that is open), A door to the living room (that is open), A door to the workshop (that is open)
**Action:** teleport to the greenhouse
**Observation:** You teleport to the greenhouse.

**Action:** connect painting to kitchen
**Observation:** terminal 1 on painting is now connected to terminal 1 on kitchen
**Action:** connect agent to bathtub
**Observation:** No known action matches that input.

Figure 6: Training Data Example for ScienceWorld.

---

Training Data Example for Game of 24

**Task Instruction:** You are a start agent and generate data for Game24. Game24 requires users to use numbers and basic arithmetic operations (+ - * /) to obtain 24.
You task is to generate a new input (4 digital number) for Game 24.
1. each new input number should be in the range of 1 to 13.
2. People can use numbers and basic arithmetic operations (+ - * /) to obtain 24. At each step, people are only allowed to choose two of the remaining numbers to obtain a new number. Here are the few-shot examples.
3. since there is only four number input and the intermediate steps should only be three.
Input: 12 10 8 4

**Positive Example:**
**Steps:** 10 - 8 = 2 (left: 2 4 12)
12 / 2 = 6 (left: 4 6)
6 * 4 = 24 (left: 24)
**Answer:** (12 / (10 - 8)) * 4 = 24

**Negative Example:**
**Steps:** 10 - 12 = -2 (left: -2 4 8)
8 / 4 = 2 (left: -2 2)
-2 * 2 = 4 (left: 4)
**Answer:** ((10 - 12) * (8 / 4)) = 4

Figure 7: Training Data Example for Game of 24.

---

Instruction Generation Prompt for ScienceWorld

**Task Instruction:** You are a helpful assistant to do some scientific experiments in an environment.

In the environment, there are several rooms: kitchen, foundry, workshop, bathroom, outside, living room, bedroom, greenhouse, art studio, and hallway.

The available actions are:
open OBJ: open a container
. . .

You will be given a dialogue between you (assistant) and a human user. You need to generate the task description after understanding the dialogue given to you. In order to help you better generate the task description, I will give you an example below.

In this example, a human user gives a Task Description at first. Then you work as an assistant and talk with user to finish the task step by step using the available actions above. Thought and Action are your reasoning and the actions you take at each step. Observation is provided by the human user; it is an observation of the surrounding environment after you have executed the action. In the end, you need to restore the Task Description based on the

whole dialogue.

**Example:**
**User**
Task Description
**Assistant**
 Thought: I need to find materials to create green paint, which are likely to be found in an art studio.
Action: look around
**User**
Observation: This room is called the bedroom. In it, you see:
...
User
Observation: Blue and yellow paint mix to produce green paint.
Assistant

Thought: The task is to create green paint and then focus on it. Now that the green paint is made, I should concentrate on the final product.
Action: focus on green paint

Task Description: Your task is to use chemistry to create green paint. When you are done, focus on the green paint.

Here is the task you need to do:
...

You should generate the Task Description following the format like examples below and generate it based on the dialogue above without any other words.
Examples:
Your task is to boil ice cream. For compounds without a boiling point, combusting the substance is also acceptable. First, focus on the substance. Then, take actions that will cause it to change its state of matter.
Your task is to use chemistry to create violet paint. When you are done, focus on the violet paint.
Your task is to find a(n) living thing. First, focus on the thing. Then, move it to the red box in the bathroom.

**Task Description:**

Figure 8: Instruction Generation Prompt for ScienceWorld.

---

**Instruction Refinement Prompt for ScienceWorld**

**Task Instruction:** You are a helpful assistant to do some scientific experiments in an environment.
In the environment, there are several rooms: kitchen, foundry, workshop, bathroom, outside, living room, bedroom, greenhouse, art studio, and hallway.
The available actions are:
open OBJ: open a container
. . .

You will be given a task description and a corresponding trajectory. The task description concludes what you have done in this trajectory. You need to elaborate this description based on this environment by adding more details.

**Example:**
**Task Description:** Your task is to grow an apple. You can find seeds in the kitchen. You should focus on the grown apple.
**Corresponding Trajectory:**
**look around**
This room is called the hallway. In it, you see:
...
**open door to kitchen**
The door is already open.
**go to kitchen**
You move to the kitchen.
...

**Refined Task Description:** Your task is to grow an apple. This will require growing several plants, and them being crosspollinated to produce fruit. Seeds can be found in the kitchen. To complete the task, focus on the grown apple.

Here is the task description you need to refine, and the corresponding trajectory is also provided:
...

**Refined Task Description:**

Figure 9: Instruction Refinement Prompt for ScienceWorld.

---

**Positive Trajectory Synthesis Prompt for ScienceWorld**

**Task Instruction:** You are a helpful assistant to do some scientific experiments in an environment.

In the environment, there are several rooms: kitchen, foundry, workshop, bathroom, outside, living room, bedroom, greenhouse, art studio, and hallway.

The available actions are:
open OBJ: open a container
. . .

Based on this environment, you need to randomly propose a Task Description, which concludes what you have done in this environment.

Here are some examples:
Your task is to use chemistry to create green paint. When you are done, focus on the green paint.
Your task is to determine whether tall plant height is a dominant or recessive trait in the pea plant. If the trait is dominant, focus on the red box. If the trait is recessive, focus on the green box.
. . .
Once you obtain the Task Description, you need to navigate through the environment to complete the instruction and generate a trajectory.

**Example:**
**Task Description:** Your task is to use chemistry to create green paint. When you are done, focus on the green paint.

**Trajectory:**
**Thought:** I need to find materials to create green paint, which are likely to be found in an art studio.
**Action:** look around
. . .

**Generated Trajectory:**

Figure 10: Positive Trajectories Synthesis Prompt for ScienceWorld.

---

Negative Trajectory Synthesis Prompt for ScienceWorld

**Task Instruction:** You are a helpful assistant to do some scientific experiments in an environment.

In the environment, there are several rooms: kitchen, foundry, workshop, bathroom, outside, living room, bedroom, greenhouse, art studio, and hallway.

The available actions are:
open OBJ: open a container
. . .

You will be given a task description and a corresponding trajectory. Based on them, you need to generate a negative sample that is similar to the correct trajectory but different from it. The generated trajectory should not meet all requirements of the task description. Moreover, the generated trajectory should satisfy all requirements of the environment.

**Example:**
**Task Description:** Your task is to focus on the life stages of the apple plant, starting from earliest to latest. The plants are located outside.

**Positive Trajectory:**
**look around**
This room is called the hallway. In it, you see:
. . .
**open door to outside**
The door is already open
. . .

**Negative Trajectory:**
**look around**
This room is called the hallway. In it, you see:
. . .
**open door to kitchen**
The door is already open.

**go to kitchen**
You move to the kitchen.
. . .

Here is the task you need to do:

...
**Negative Trajectory:**

---

Figure 11: Negative Trajectories Synthesis Prompt for ScienceWorld.

**Reward Model Training Details.** The detailed hyperparameters we use for reward model during training and inference are shown in Table 14. We employ identical hyperparameters for reward models of different environments. For Webshop, we use checkpoint of 1100 steps in ARMAP-B, and checkpoint of 1200 steps in ARMAP-R and ARMAP-M.

| Name | ScienceWorld | Webshop | Game of 24 |
|---|---|---|---|
| lora r | | 64 | |
| lora alpha | | 16 | |
| lora dropout | | 0.0 | |
| lora target modules | q_proj, k_proj, v_proj, o_proj, gate_proj, up_proj, down_proj | | |
| epochs | 10 | 3 | 10 |
| batch size | 8 | 1 | 4 |
| batch size per device | 1 | 1 | 1 |
| gradient accumulation steps | 16 | 4 | 16 |
| learning rate | 1e-5 | 2e-5 | 1e-5 |
| warmup ratio | 0.2 | 0.1 | 0.25 |
| checkpoint steps | 160 | 1100, 1200 | 1500 |
| temperature | 0.0 | 0.0 | 0.0 |

Table 14: Detailed hyperparameters used in reward model.

**Implementation Details of Ablation baselines.** For SFT, we use all positive examples from the reward model training as the training data. The training objective is to enable the model to predict the output of the LLM in the positive examples.

For using few-shot prompting to guide the LLMs to predict the reward of historical trajectories, we use the following format of the few-shot prompt:

---

**Few-shot Prompt for LLMs Directly Serving as ScienceWorld Reward Model**

**Task Instruction:** You are an autonomous intelligent agent tasked with evaluating the trajectories of the past experience. You will be given the history of a past experience in which you were placed in an environment and given a task to complete. These tasks will be accomplished through the use of specific actions. Now you are trying to evaluate the performance on a past task. You will be given the objective of the task, the history of interaction including the observations you had and the actions you issued, and the status of the task. Your goal is to think about the strategy and provided path to produce a score ranging from 0 to 1 to measure whether the objective of the task has been reached.

Here are 2 examples:

**Example1:**

**Human:** You are a helpful assistant to do some scientific experiment in an environment. In the environment, there are several rooms: kitchen, foundry, workshop, bathroom, outside, living room, bedroom, greenhouse, art studio, hallway. You should explore the environment and find the items you need to complete the experiment. You can teleport to any room in one step. All containers in the environment have already been opened, you can directly get items from the containers. The available actions are: open OBJ: open a container, close OBJ: close a container, activate OBJ: activate a device, deactivate OBJ: deactivate a device, connect OBJ to OBJ: connect electrical components, disconnect OBJ: disconnect electrical components, use OBJ [on OBJ]: use a device/item, look around: describe the

---

current room, examine OBJ: describe an object in detail, look at OBJ: describe a container's contents, read OBJ: read a note or book, move OBJ to OBJ: move an object to a container, pick up OBJ: move an object to the inventory, pour OBJ into OBJ: pour a liquid into a container, mix OBJ: chemically mix a container, teleport to LOC: teleport to a specific room, focus on OBJ: signal intent on a task object, wait: task no action for 10 steps, wait1: task no action for a step

**Task Description:** Your task is to find a(n) non-living thing. First, focus on the thing. Then, move it to the purple box in the living room. Please only output one Thought and one Action. The available actions are: open OBJ: open a container, close OBJ: close a container, activate OBJ: activate a device, deactivate OBJ: deactivate a device, connect OBJ to OBJ: connect electrical components, disconnect OBJ: disconnect electrical components, use OBJ [on OBJ]: use a device/item, look around: describe the current room, examine OBJ: describe an object in detail, look at OBJ: describe a container's contents, read OBJ: read a note or book, move OBJ to OBJ: move an object to a container, pick up OBJ: move an object to the inventory, pour OBJ into OBJ: pour a liquid into a container, mix OBJ: chemically mix a container, teleport to LOC: teleport to a specific room, focus on OBJ: signal intent on a task object, wait: task no action for 10 steps, wait1: task no action for a step. Please strictly follow the format of the actions.

**Assistant:** Thought: I need to find a non-living thing, which could be an object in any room, so I should start by looking around the current room. Action: look around

**Human:** Observation: This room is called the bathroom. In it, you see: the agent, a substance called air, a bathtub, which is turned off. In the bathtub is: nothing. a drawing, a glass cup (containing nothing), a sink, which is turned off. In the sink is: nothing. a toilet. In the toilet is: A drain, which is open, a substance called water. You also see: A door to the kitchen (that is open). Please only output one Thought and one Action. The available actions are: open OBJ: open a container, close OBJ: close a container, activate OBJ: activate a device, deactivate OBJ: deactivate a device, connect OBJ to OBJ: connect electrical components, disconnect OBJ: disconnect electrical components, use OBJ [on OBJ]: use a device/item, look around: describe the current room, examine OBJ: describe an object in detail, look at OBJ: describe a container's contents, read OBJ: read a note or book, move OBJ to OBJ: move an object to a container, pick up OBJ: move an object to the inventory, pour OBJ into OBJ: pour a liquid into a container, mix OBJ: chemically mix a container, teleport to LOC: teleport to a specific room, focus on OBJ: signal intent on a task object, wait: task no action for 10 steps, wait1: task no action for a step. Please strictly follow the format of the actions.

**Assistant:** Thought: The glass cup in the bathroom is a non-living thing, so I should focus on it. Action: focus on glass cup.

...

**Assistant:** Thought: I've found the purple box, and now I need to move the glass cup into it. Action: move glass cup to purple box.

**Human:** Observation: No known action matches that input.

**Human:** Please evaluate whether you complete the 'Task Description' or not.

Following your definitions, the task completion score is <0.750>

**Example2:** ...

Now is your turn:

Figure 12: Few-shot Prompt for LLMs Directly Serving as ScienceWorld Reward Model.

## A.4 MORE QUALITATIVE RESULTS.

We show more qualitative results of our ARMAP in Fig. 13 and Fig. 14. Based on the examples, we can see that the automatic reward models in our pipeline can distinguish good trajectories from bad ones and guide LLMs to generate better trajectories to finish the tasks.

---

**Qualitative Visualization for ScienceWorld**

**Task Description:** Task Description: Your task is to find a(n) non-living thing. First, focus on the thing. Then, move it to the orange box in the living room.

**Trajectory A:**

**Observation:** You teleport to the living room.

**Thought:** Now that I'm in the living room, I should look around to find a non-living thing that I can move to the orange box. **Action:** look around

**Observation:** This room is called the living room. In it, you see:
the agent, a substance called air, a chair. On the chair is: nothing. a couch. On the couch is: a white pillow. a finger painting a orange box (containing nothing) a table. On the table is: nothing.
You also see: A door to the hallway (that is open).

**Thought:** I see several objects in the living room, but I need to find a non-living thing that I can move to the orange box. **Action:** examine white pillow

ARMAP **Aborted**

**Trajectory B:**

**Observation:** This room is called the workshop. In it, you see:
the agent, a substance called air, a table. On the table is: a battery, a blue wire, a orange light bulb, which is off, a red light bulb, which is off, a red wire, a switch, which is off, a yellow light bulb, which is off, a yellow wire. a ultra low temperature freezer. The ultra low temperature freezer door is open. In the ultra low temperature freezer is: nothing. You also see:
A door to the hallway (that is open)

**Thought:** I see several objects in the workshop that could be considered non-living things, such as the battery, blue wire, or light bulbs. I should focus on one of these objects to complete the task. **Action:** focus on battery

**Observation:** You focus on the battery.

**Thought:** Now that I've focused on the battery, I need to move it to the orange box in the living room. **Action:** teleport to living room

**Observation:** You teleport to the living room.

**Thought:** Now that I'm in the living room, I should look around to find the orange box and move the battery into it. **Action:** look around

**Observation:** This room is called the living room. In it, you see:
the agent, a substance called air, a chair. On the chair is: nothing. a couch. On the couch is: a white pillow. a finger painting a orange box (containing nothing) a table. On the table is: nothing.
You also see: A door to the hallway (that is open)

**Thought:** I've found the orange box in the living room, and now I can move the battery into it. **Action:** move battery to orange box

ARMAP **Selected**

Figure 13: Qualitative Visualization for ScienceWorld. Based on the comparison of the two trajectories, it is evident that the final action of trajectory B brings the resulting state closer to the task objective. Therefore, our model identifies trajectory B as the superior option.

---

**Qualitative Visualization for Game of 24**

**Input A:** 2 5 8 11
**Trajectory A1:**
**Steps:** 8 * 2 = 16 (left: 5 11 16)
11 + 5 = 16 (left: 16 16)
16 + 16 = 32 (not 24, not valid)
11 - 5 = 6 (left: 2 6 8)
8 / 2 = 4 (left: 4 6)
4 * 6 = 24 (left: 24)
**Answer:** (8 / 2) * (11 - 5) = 24
ARMAP **Selected**

**Trajectory A2:**
**Steps:**
8 - 5 = 3 (left: 2 3 11)
11 - 3 = 8 (left: 2 8)
2 * 8 = 16 (left: 16)
16 + 8 = 24 (left: 24)
**Answer:** (((8 - 5) - 8) * 2) + 8 = 24
ARMAP **Aborted**

**Input B:** 2 3 6 9
**Trajectory B1:**
**Steps:** 9 - 3 = 6 (left: 2 6 6)
6 + 6 = 12 (left: 2 12)
2 * 12 = 24 (left: 24)
**Answer:** 2 * (6 + (9 - 3)) = 24
ARMAP **Selected**

**Trajectory B2:**
**Steps:**
9 - 3 = 6 (left: 2 6 6)
6 - 2 = 4 (left: 4 6)
6 * 4 = 24 (left: 24)
**Answer:** (6 * (9 - 3)) = 24
ARMAP **Aborted**

---

Figure 14: Qualitative Visualization for Game of 24. Trajectory A and Trajectory B correspond to input A and input B respectively. Results show that our ARMAP can accurately pick out the correct trajectory.

## A.5 FAILURE CASE ANALYSIS

In this section, we investigate the common failure cases of our framework, aiming to provide data points and insights for future research.

The most common error occurs when there are multiple restrictions in the instruction, the reward model overlooks some of these key conditions. A representative example is illustrated in Fig. 15, the model focuses on price and size but ignores the details about 'Fluoride' hidden in the product description.

Another common failure mode occurs when commonsense knowledge is involved. As demonstrated in Fig. 16, the agent was tasked with buying a blackout shade but failed to choose both the color and the size. While, in everyday life, size is generally more important, the reward model prioritized color instead. In Fig. 17, the reward model cannot assess the lifespan of dragonflies and chipmunks because it lacks the necessary biological knowledge.

**Discussion.** The analysis of failure modes highlights the significant potential of our framework. To improve its performance, we propose two possible strategies for improvements in reward modeling: **(a)** Constructing Data with Focus on Complex and Detailed Conditions: enhancing the dataset to include scenarios with higher complexity and more nuanced conditions will help the framework better handle intricate situations and edge cases. **(b)** Intervening in Reward Scoring with External Knowledge: incorporating external knowledge by combining a prompted Large Language Model with the trained reward model. This approach allows the LLM's generalized knowledge to calibrate the reward scores in a controllable manner, improving the overall accuracy and robustness of the reward model.

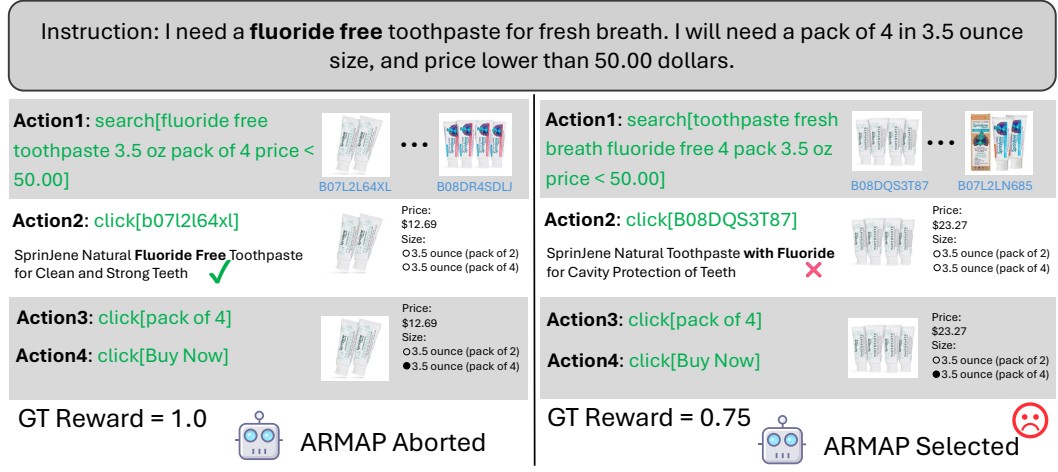

Figure 15: **Failure Example from Webshop**. The reward model ignores certain key conditions in the task instruction.

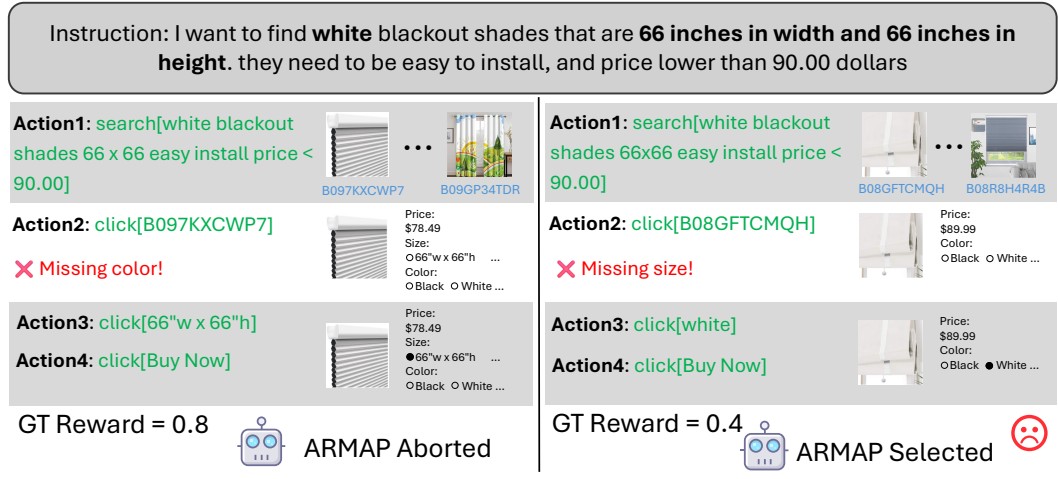

Figure 16: **Failure Example from Webshop**. The reward model misjudged the importance of different conditions, such as the size and color in this case.

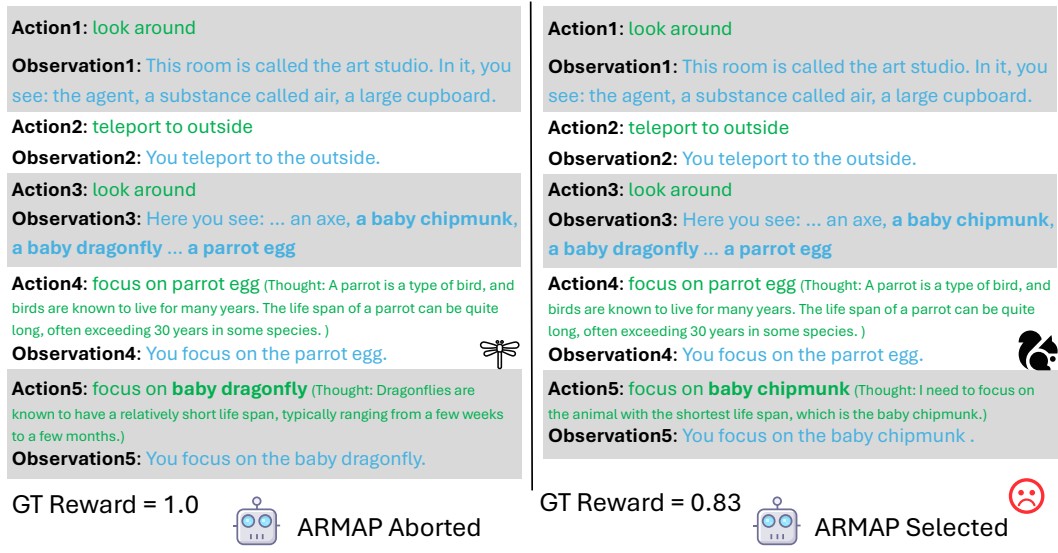

Figure 17: **Failure Example from ScienceWorld**. Reward models have problems when common-sense knowledge is involved, such as the inability to determine the shortest life span.

