# OpenReview forum: "Scaling Autonomous Agents via Automatic Reward Modeling And Planning"
_ICLR.cc/2025/Conference — ICLR 2025 Poster_

### Official Review · Reviewer_Rvsi · 2024-11-03

**Soundness:** 4
**Presentation:** 3
**Contribution:** 3
**Rating:** 8
**Confidence:** 4

**Summary:**

ARMAP presents a novel framework for autonomous agents by leveraging reward modeling and planning. It trains a reward model on contrastive trajectories, enabling effective decision-making in complex environments through LLM-as-agents. Unlike input-optimized prompting-based approaches, ARMAP scores steps within task trajectories, focusing on task completion. The ablation study supports the framework’s effectiveness and adaptability.

**Strengths:**

Originality: The automatic reward model and data generation approach presented is novel, allowing the framework to guide task completion within complex decision-making environments effectively.

Quality: ARMAP stands out by using a reward model to evaluate and guide navigation steps in agentic environments, enhancing decision-making processes and setting a solid foundation for handling intricate tasks autonomously.

Clarity: The paper is well-written, with a clear flow that effectively communicates the core concepts and approach. While a few notational details could be clarified, the overall presentation is strong and accessible.

Significance: The framework's value is demonstrated through LLM-agent task performance, highlighting flexibility in controllable task generation and practical application via a reward model, which reduces reliance on large LLMs or human labeling.

**Weaknesses:**

Specificity in Reward Model Design: The paper lacks detailed information on the size and neural architecture of the reward model. Additionally, challenges in reward model development are not clearly defined. More depth and specific examples are needed to clarify these choices and support the framework's claims.

Limited Dataset Scope: The study could benefit from evaluating on a broader set of complex, long-trajectory decision-making agent datasets. Including established datasets such as AlfWorld or BabyAGI, which could strengthen the empirical evaluation and demonstrate robustness across diverse environments.

Insufficient Detail on Multimodal and Visual Input Integration: While the paper mentions multimodal feedback and visual inputs, it lacks clarity of their impact on reward model training. An ablation study that isolates the effect of visual inputs compared to text-based inputs could better illustrate their importance and further validate the framework’s design.

**Questions:**

Although the automatic reward model training is a good idea, there are few concerns after going through the paper and demand clarity of choice:
1. Writing and Formatting:
    * In Figure 1, the title "Tree Planning" should use lowercase "(c)" instead of capital "(C)."
2. Reward Model Specifics:
    * Could authors clarify the size of the reward model used in this study?
    * In Line 100, authors mention challenges in developing a reward model (RM). Could they provide a few specific examples of these challenges for clarity?
    * What neural architecture was selected for the reward model in this framework? Is this inspired from any previous works?
3. Dataset Selection:
    * Some established decision-making agent datasets, such as AlfWorld, BabyAGI, or PDDL, are not included. These embodied agent datasets offer complex, long trajectories that could be valuable to the study. Could authors comment on their absence or suitability?
4. Multimodal Feedback:
    * Line 150 refers to multimodal feedback. Could you specify which modalities other than text were used in predicting the next action?
5. Reward Model Type:
    * In Line 161, you state a focus on developing the reward model. Is this a classification model with a defined set of output classes, or is it a regression model?
6. Observation Clarification:
    * In Line 225, the phrase “...corresponding environment observations...” could benefit from refinement, as there’s typically one extra observation at the start. Could this section be adjusted to clarify the distinction?
7. Trajectory Generation and Instruction Use:
    * In Figure 2, authors mention using “initial language instructions in the environment” to generate trajectories, but it’s unclear if any LLM was employed to identify keywords. For instance, in “I am looking for jeans with 40w x 34l size, and price lower than 200 dollars,” did the framework use LLM predictions to determine "Jeans" as the keyword for search?
8. Impact of Visual Inputs:
    * What role do visual inputs play in the reward model’s training? Have authors conducted any ablation studies that use only text from trajectories to measure their impact? It would be helpful to know if the visual inputs significantly influence the final model performance. I find this missing.

These points would enhance the clarity and depth of the paper, particularly around architectural choices and empirical coverage. I am looking forward to the rebuttal during the discussion phase.

---

> ### Author Response · Authors · 2024-11-25
> **Look forward for your constructive response (Part 1)**
>
> **Q1. Specificity of Reward Model Design.**
> > Specificity in Reward Model Design: The paper lacks detailed information on the size and neural architecture of the reward model. Additionally, challenges in reward model development are not clearly defined. More depth and specific examples are needed to clarify these choices and support the framework's claims.
>
> During the rebuttal phase, we further study the size and type of the reward modelings. We show the performance of using different model types and sizes in Table 3-1 and Table 3-2.
>
> | ARMAP-B  | VILA-3B    | VILA-13B   | LLaVA-13B
> | -------- | ----- | ----- | -----
> | LLaMA-70B | 57.3% | 61.2% | 44.3%
> | LLaMA-8B  | 35.7% | 34.3% | 26.0%
> | Mistral-7B  | 24.5% | 26.0% | 19.5%
> | Phi-3.8B      | 20.0% | 19.5% | 16.7%
>
> Table 3-1: The effects of our ARMAP-B method under different policy models in the context of varying reward models types and sizes in ScienceWorld Seen.
>
> | ARMAP-B  | VILA-3B    | VILA-13B   | LLaVA-13B
> | -------- | ----- | ----- | -----
> | LLaMA-70B | 57.0% | 60.7% | 48.2%
> | LLaMA-8B  | 28.1% | 27.5% | 22.2%
> | Mistral-7B  | 21.1% | 22.9% | 19.2%
> | Phi-3.8B      | 17.0% | 15.3% | 13.7%
>
> Table 3-2: The effects of our ARMAP-B method under different policy models in the context of varying reward models types and sizes in ScienceWorld Unseen.
>
> We utilize VILA-3B as our reward model in the paper. VILA is the improved version based on LLaVA, which is trained on more interleaved image-text data and achieves better performance. Moreover, it has a well-maintained open-sourced codebase. Different from previous works using commercial and private models like GPT series, we utilize the free and open-sourced LLMs. It is acknowledged that the open-sourced small-scale models is relatively less performant than larger-scale models, so we need to spend more time on dataset creation and planning algorithms to make sure that even if we employ small models, we can also obtain good results. More details can be found at **Q5**.
>
> In Tables 3-1 and 3-2, we observe that using VILA leads to better performance. Moreover, by further enhancing VILA, we can achieve even better results. Considering the demand for efficiency and resources in practical applications, we opt for the VILA-3B model. However, if better performance is required, we can also employ a larger reward model.
>
> **Q2. Larger Dataset Scope.**
> > Limited Dataset Scope: The study could benefit from evaluating on a broader set of complex, long-trajectory decision-making agent datasets. Including established datasets such as AlfWorld or BabyAGI, which could strengthen the empirical evaluation and demonstrate robustness across diverse environments.
>
> We thank the reviewers for suggestions on more experiments in new and more diverse environments. During rebuttal, we have added new experiments in ALFWorld [1] and ClinicalAgent [2].
>
> **Effectiveness of Our Pipeline.** We conduct more experiments in different tasks across two different domain in **G2**. Please refer to **G2** for more analysis and details.
>
> As for BabyAGI [3], we find it is more like a framework for tasks scheduling rather than a standard benchmark or environment for LLM agents, which is far from the scope of our framework. Thus, we leave it as future work.
>
> [1] Mohit Shridhar, et al. Alfworld: Aligning text and embodied environments for interactive learning. In International Conference on Learning Representations, 2020.
>
> [2] S. Schmidgall, R. Ziaei, C. Harris, E. Reis, J. Jopling, and M. Moor, “Agentclinic: a multimodal agent benchmark to evaluate AI in simulated clinical environments,” 2024.
>
> [3] BabyAGI: https://github.com/yoheinakajima/babyagi

---

> > ### Author Response · Authors · 2024-11-25
> > **Look forward for your constructive response (Part 2)**
> >
> > **Q3. Ablation on Visual Input.**
> > > Insufficient Detail on Multimodal and Visual Input Integration: While the paper mentions multimodal feedback and visual inputs, it lacks clarity of their impact on reward model training. An ablation study that isolates the effect of visual inputs compared to text-based inputs could better illustrate their importance and further validate the framework’s design.
> > > What role do visual inputs play in the reward model’s training? Have authors conducted any ablation studies that use only text from trajectories to measure their impact? It would be helpful to know if the visual inputs significantly influence the final model performance. I find this missing.
> >
> > Thanks for the suggestion to add ablation to study the effectiveness of visual context on Webshop. During rebuttal, we have trained a new reward model without visual information. As shown in Table 3-3 and 3-4, we can see that, in different settings, the reward model with visual information performs better than the model without visual information, which shows the value of visual context for the Webshop task. We also want to highlight that the main contribution of our paper is not the integration of visual information but to introduce a novel framework ARMAP for LLM-based agents incorporating an automatic reward model and different planning algorithms for complex agent task solving.
> >
> > |ARMAP-B |w/o Visual|w/ Visual|
> > |--------|--------|--------|
> > |LLaMA-70B|61.6%|62.0%
> > |Mistral-7B|51.3%|54.4%
> >
> > Table 3-3. Ablation of the visual input on ARMAP-B in Webshop task.
> >
> > |ARMAP-R |w/o Visual|w/ Visual|
> > |--------|--------|--------|
> > |LLaMA-70B|56.1%|56.5%
> > |Mistral-7B|53.6%|54.1%
> >
> > Table 3-4. Ablation of the visual input on ARMAP-R in Webshop task.
> >
> > **Q4. Writing and Formatting.**
> > >In Figure 1, the title "Tree Planning" should use lowercase "(c)" instead of capital "(C)."
> >
> > Thanks for your correction and we will revise the paper accordingly.
> >
> > **Q5. More Details about Reward Model**
> > >Could authors clarify the size of the reward model used in this study?
> > In Line 100, authors mention challenges in developing a reward model (RM). Could they provide a few specific examples of these challenges for clarity?
> > What neural architecture was selected for the reward model in this framework? Is this inspired from any previous works?
> >
> > **Challenges.** Previously, people used powerful commercial LLM APIs to evaluate different tasks, which are expensive and hard to scale up. Moreover, previous works didn't consider using and integrating various planning algorithms for problem-solving. In contrast, we propose an automated approach to generate data and learn a multi-modal reward models. In addition, we only employ the open-sourced LLMs to do experiments.
> >
> > **About the model we choose.** We utilize the VILA-3B model as our reward model in the paper. This model is an improvement of LLaVA and it is trained on additional interleaved image-text data which enables the stronger multi-modal understanding ability. We choose this model not just becuase of its strong performance, but also becuase they maintain a good codebase and detailed technical document. During the rebuttal phase, we also add LLaVA-3B and LLaVA-13B for comparison.
> >
> > **Q6. Dataset Selection**
> > > Some established decision-making agent datasets, such as AlfWorld, BabyAGI, or PDDL, are not included. These embodied agent datasets offer complex, long trajectories that could be valuable to the study. Could authors comment on their absence or suitability?
> >
> > Thanks for the suggestion to conduct experiments on more agent tasks. During the rebuttal, we have added the ALFWorld and ClinicalAgent for experiments. Please refer to **G2** in General Response for more details and analysis.
> >
> > **Absence and suitability of PDDL[1] and BabyAGI[2].** PDDL is Planning Domain Definition Language, which is used by ALFWorld to describe each scene from ALFRED[3] and to construct an equivalent text game using the TextWorld[4] engine. The dynamics of each game are defined by the PDDL domain. Therefore, we consider PDDL to be a language for describing processes, and it is not suitable for our research tasks. For BabyAGI, they are not naturally suitable for evaluating the performance of our framework since BabyAGI is a framework ratehr tha a benchmark for LLM task scheduling. We might extend our ARMAP pipeline to utilize these libraries or write it in PDDL in the future.
> >
> > [1] D. McDermott, et al. PDDL: the planning domain definition language.
> >
> > [2] BabyAGI: https://github.com/yoheinakajima/babyagi
> >
> > [3] Mohit Shridhar, et al. ALFRED: A Benchmark for Interpreting Grounded Instructions for Everyday Tasks. CVPR 2020.
> >
> > [4] Marc-Alexandre Côté, et al. TextWorld: A Learning Environment for Text-based Games.

---

> > > ### Author Response · Authors · 2024-11-25
> > > **Look forward for your constructive response (Part 3)**
> > >
> > > **Q7. Multimodal Feedback.**
> > > > Line 150 refers to multimodal feedback. Could you specify which modalities other than text were used in predicting the next action?
> > >
> > > The multimodal feedback in Webshop tasks returns the updated website as feedbacks, which we represent the websites as an HTML text file and a screenshot for visual perception. For other environments, the feedback is text-based. We also conduct experiments to see the effect of visual input, please also refer to **Q3**.
> > >
> > > Q8. **Reward Model Type.**
> > > > In Line 161, you state a focus on developing the reward model. Is this a classification model with a defined set of output classes, or is it a regression model?
> > >
> > > It is a regression model that output a normalized value ranging from 0 to 1 to predict the reward value of the trajectory. This formulation is widely used in reward modeling, which can be seen in [1,2,3,4,5].
> > >
> > > [1]. Bradley, Ralph Allan, and Milton E. Terry. "Rank analysis of incomplete block designs: I. The method of paired comparisons." Biometrika 39.3/4 (1952): 324-345.
> > >
> > > [2]. Touvron, Hugo, et al. "Llama 2: Open foundation and fine-tuned chat models." arXiv preprint arXiv:2307.09288 (2023).
> > >
> > > [3]. Dubey, Abhimanyu, et al. "The llama 3 herd of models." arXiv preprint arXiv:2407.21783 (2024).
> > >
> > > [4]. Yang, An, et al. "Qwen2 technical report." arXiv preprint arXiv:2407.10671 (2024).
> > >
> > > [5]. Sun, Zhiqing, et al. "Principle-driven self-alignment of language models from scratch with minimal human supervision." Advances in Neural Information Processing Systems 36 (2024).
> > >
> > > **Q9. Observation Clarification.**
> > > > In Line 225, the phrase “...corresponding environment observations...” could benefit from refinement, as there’s typically one extra observation at the start. Could this section be adjusted to clarify the distinction?
> > >
> > > Thanks for the suggestions to clarify the notation of the observations and actions. Since we always start by the intial observation of the system, which we represent it as $o_0$. And then we take the action $a_1$ and the corresponding updated observation $o_1$. We repeat the process of taking the action and get the observation for $N$ times. Together we have $N+1$ observations (*i.e.*  $\{o_n\} _ {n=0}^N$ ) and $N$ actions (*i.e.*  $\{a_n\}_{n=1}^N$ ). We have added this explanation in the revised paper.
> > >
> > > **Q10. Trajectory Generation and Instruction Use.**
> > > > In Figure 2, authors mention using “initial language instructions in the environment” to generate trajectories, but it’s unclear if any LLM was employed to identify keywords. For instance, in “I am looking for jeans with 40w x 34l size, and price lower than 200 dollars,” did the framework use LLM predictions to determine "Jeans" as the keyword for search?
> > >
> > > Thanks for the suggestion to clarify the details of instruction matching. We do **not** use any LLM to identify the keyword for search. We simply use the search engine in the Webshop environment for matching instructions with title, description, overview, and customization options of the websites. In the Webshop environment, it uses Pyserini[1] as the search engine for shopping websites. Please refer to the Webshop paper [2] for more details.
> > >
> > > [1]. Lin, Jimmy, et al. "Pyserini: An easy-to-use python toolkit to support replicable ir research with sparse and dense representations." arXiv 2021.
> > >
> > > [2]. Yao, Shunyu, et al. "Webshop: Towards scalable real-world web interaction with grounded language agents." NeurIPS 2022.

---

> > > > ### Comment · Reviewer_Rvsi · 2024-11-26
> > > > **Key Clarifications and Suggested Additions**
> > > >
> > > > Dear Authors, From the given responses
> > > > * From response to Q1 having Table 3-1 and 3-2, performance varies when using different sized reward models and thanks for clarifying this aspect with additional numbers.
> > > > * Response to Q3 clarifies to me the effect of w and w/o visual inputs to the model, which must be added to final draft.
> > > > * Response to Q5, mentioning “....we only employ the open-sourced LLMs to do experiments” could be added as a limitation of this work with focus where an upper bound of performance with API based LLMs can be calculated.
> > > > * In response to Q10, then in Figure 2, left most side, “search[jeans]” should be replaced with instruction showcasing as “search[…jeans…]”
> > > >
> > > > I keep my score unchanged for now. Thankyou for additional information.

---

> > > > > ### Author Response · Authors · 2024-11-28
> > > > > **Thanks for your constructive comments**
> > > > >
> > > > > Thanks again for your constructive response.
> > > > >
> > > > > >Response to Q3 clarifies to me the effect of w and w/o visual inputs to the model, which must be added to final draft.
> > > > >
> > > > > We have added this part to the revised paper we uploaded.
> > > > >
> > > > > >Response to Q5, mentioning “....we only employ the open-sourced LLMs to do experiments” could be added as a limitation of this work with focus where an upper bound of performance with API based LLMs can be calculated.
> > > > >
> > > > > |GPT-4o |Std|Dev
> > > > > |--------|--------|--------|
> > > > > |Sampling|0.74|0.88
> > > > > |Greedy|0.82|0.90
> > > > > |ARMAP-B | 0.84|0.95
> > > > >
> > > > > Table 3-5. New experiments by using the API based LLMs on ALFWorld
> > > > >
> > > > > To serve as the training data generator, closed-source models have several drawbacks, including high costs, limited commercial access, and lack of reproducibility. In contrast, our approach achieves strong results without relying on closed-source models. Given the expense associated with API-based models like GPT-4o for generating training datasets, we have opted not to pursue this method for now.
> > > > >
> > > > > For API-based models serving as policy models, the high cost of GPT-4o and API access rate limitations prompted us to focus our experiments primarily on ALFWorld. Specifically, we used GPT-4o-2024-08-06 to sample five trajectories each on ALFWorld’s Dev and Std sets, then conducted experiments using our automatic reward model. As shown in the table above,  our reward model is able to help the powerful GPT-4o gain better performance, demonstrating the effectiveness of our framework.
> > > > >
> > > > > We also added this part in our revised paper.
> > > > >
> > > > > >In response to Q10, then in Figure 2, left most side, “search[jeans]” should be replaced with instruction showcasing as “search[…jeans…]”
> > > > >
> > > > > We have revised this part. Please check the latest version we uploaded.

---

> > > > > > ### Comment · Reviewer_Rvsi · 2024-11-28
> > > > > > **Comprehensive study**
> > > > > >
> > > > > > Thank you for providing the additional data and conducting the ablation with GPT-4o. The paper demonstrates a comprehensive and in-depth exploration of the study, addressing key aspects effectively. Therefore, I update my rating for this work.

---

> > > > > > > ### Author Response · Authors · 2024-11-28
> > > > > > > **Thanks again for your detailed response**
> > > > > > >
> > > > > > > Thanks again for your detailed and constructive response. We are glad you recognize the merit of the paper!

---

### Official Review · Reviewer_XSp2 · 2024-11-06

**Soundness:** 2
**Presentation:** 2
**Contribution:** 2
**Rating:** 6
**Confidence:** 3

**Summary:**

The paper proposes a framework named ARMAP, aimed at enhancing the task-solving capabilities of LLM-based agents in challenging environments that necessitate multi-step decision-making. While traditional LLMs perform well in text-based tasks, they face challenges with interactive, goal-oriented tasks due to limited access to large-scale decision-making data. ARMAP tackles these issues by developing an automated reward model that assesses action trajectories without requiring human annotations.

The framework comprises three main components:
1. Data Generation: An LLM agent interacts with the environment, producing diverse action trajectories that include both successful and unsuccessful task completion attempts. These trajectories, encompassing task intents, positive outcomes, and negative outcomes, are utilized to train the reward model.
2. Reward Model: A specialized model evaluates the effectiveness of each trajectory in fulfilling a task, thereby guiding the LLM agents in their planning.
3. Planning Algorithms: By integrating the reward model with planning methods like Monte Carlo Tree Search (MCTS) and Reflexion, the agent can optimize its actions to follow high-reward paths.

Experiments depict ARMAP’s efficacy across various benchmarks, demonstrating improved planning performance for different LLM agents. The approach offers advantages in flexibility and practicality, as it reduces reliance on human labels and expensive, closed LLMs, thereby facilitating the development of more autonomous and efficient AI agents capable of managing real-world tasks.

**Strengths:**

Automated Reward Modeling: It presents an innovative method for autonomously learning reward models without the need for human-annotated data, addressing issues related to data scarcity and dependence on costly closed-source LLMs. This makes the framework scalable and practical for real-world applications.

Enhanced Decision-Making for LLM Agents: By offering a reward-based evaluation system, ARMAP significantly boosts the ability of LLM agents to perform complex, multi-step tasks that require sequential planning, an area where standard LLMs often struggle.

Efficiency and Cost-Effectiveness: By eliminating the need to fine-tune LLMs and avoiding reliance on proprietary LLM APIs, ARMAP provides a cost-effective solution that could make high-performing AI agents more accessible for widespread use.

**Weaknesses:**

Limited Applicability in Highly Dynamic Environments: While the framework performs well in simulated environments with fixed rules, such as online shopping simulations and controlled benchmarks, its effectiveness in rapidly changing, unpredictable real-world environments is uncertain. The model may struggle with scenarios that require quick adaptation to new patterns not present in the training data.

Computational Overhead with Complex Planning: The integration of planning algorithms like MCTS, while effective, can introduce significant computational costs, especially when exploring multiple trajectories. This may limit ARMAP’s efficiency in resource-constrained settings or for tasks requiring real-time responses.

**Questions:**

Synthetic Data Quality: How do you ensure the quality and diversity of the synthetic trajectories generated by LLMs? Have you observed any limitations when these synthetic trajectories don’t align closely with real-world decision-making patterns?

Computational Cost in Real-Time Applications: Given the computational demands of planning algorithms like MCTS, how would ARMAP perform in applications requiring real-time decision-making? Are there strategies for reducing overhead while retaining performance?

Reward Model Generalization: How well does the reward model generalize to tasks and environments different from those it was trained on? Have you tested ARMAP in domains requiring more complex, domain-specific knowledge, such as legal or medical contexts?

Scalability and Practical Deployment: What are the main challenges you foresee in scaling ARMAP for broader deployment in real-world applications? Are there specific areas (e.g., hardware requirements, integration with other models) that need further development?

---

> ### Author Response · Authors · 2024-11-25
> **Look forward for your further feedback (Part 1)**
>
> **Q1. More Environments for Experiments.**
> > Limited Applicability in Highly Dynamic Environments: While the framework performs well in simulated environments with fixed rules, such as online shopping simulations and controlled benchmarks, its effectiveness in rapidly changing, unpredictable real-world environments is uncertain. The model may struggle with scenarios that require quick adaptation to new patterns not present in the training data.
>
> We thank the reviewers for their thoughtful suggestion to evaluate ARMAP in more diverse and high-stakes environments, as well as their observations regarding its applicability to dynamic, unpredictable scenarios. To address these concerns, we have conducted additional experiments during the rebuttal phase, expanding our evaluation to ALFWorld [1] and ClinicalAgent [2], which demand advanced reasoning, adaptability, and goal alignment.
>
> **Experiments in New Environments.** We provide more details about the new experiments of new environments in **G2**. Please refer to **G2** for more analysis.
>
> **Performance Adaption in New Environments.** Our results indicate that ARMAP demonstrates some capability to adapt to unseen patterns not present in the training data. For example, in ScienceWorld and ALFWorld, ARMAP performs effectively on the unseen split, which introduces new tasks and patterns not encountered during training.
> However, we do acknowledge the limitations of our framework. While ARMAP performs well in many settings and environments, building a **general reward model** capable of handling **all scenarios** with a single framework remains an open challenge. We regard this as an important direction for future research and a necessary step toward the development of more robust, universally adaptable reward models.
>
> While computational overhead is a valid concern, the robustness and adaptability provided by planning-based approaches like ARMAP make them highly effective in high-stakes and complex reasoning scenarios. By leveraging insights from test-time scaling laws and exploring targeted optimization strategies, ARMAP aims to strike a balance between computational efficiency and effectiveness, ensuring its applicability across a wide range of domains. We appreciate the reviewer’s feedback, which has helped us clarify this trade-off and outline potential paths for future work. We add such discussion in the appendix during rebuttal.
>
> [1] Mohit Shridhar, et al. Alfworld: Aligning text and embodied environments for interactive learning. In International Conference on Learning Representations, 2020.
>
> [2] S. Schmidgall, R. Ziaei, C. Harris, E. Reis, J. Jopling, and M. Moor, “Agentclinic: a multimodal agent benchmark to evaluate AI in simulated clinical environments,” 2024.

---

> > ### Author Response · Authors · 2024-11-25
> > **Look forward for your further feedback (Part 2)**
> >
> > **Q2. Computation Overhead.**
> > > Computational Overhead with Complex Planning: The integration of planning algorithms like MCTS, while effective, can introduce significant computational costs, especially when exploring multiple trajectories. This may limit ARMAP’s efficiency in resource-constrained settings or for tasks requiring real-time responses.
> >
> > **Efficiency vs. Effectiveness in the ARMAP Framework.** We thank the reviewers for highlighting this critical point about computational overhead. While we acknowledge that planning algorithms such as MCTS can introduce significant computational demands, particularly in resource-constrained settings or tasks requiring real-time responses, we believe that the effectiveness and robustness provided by planning can justify these costs in specific high-stakes scenarios.
> >
> > **Effectiveness in High-Stakes Scenarios.** Planning algorithms like MCTS excel in scenarios where the consequences of suboptimal decisions are severe, and the ability to explore multiple trajectories and reason about future outcomes is essential. For instance,
> > * Healthcare: Accurate, well-considered decisions can directly impact patient outcomes.
> > * Robotics: Tasks often require precise long-horizon planning to ensure safety and success in dynamic environments.
> > * Mathematics and Scientific Problem-Solving: These domains benefit from exploring complex reasoning pathways to arrive at optimal solutions.
> >
> > In these cases, simpler heuristic-driven methods may lack the robustness and adaptability necessary for achieving high-quality outcomes, making planning-based approaches like ARMAP invaluable despite their computational overhead.
> >
> > **Insights from Test-Time Scaling Laws.** The trade-off between effectiveness and efficiency is particularly well-illustrated by test-time scaling laws observed in large-scale AI systems. The OpenAI Open-O1 model [1, 2], for example, demonstrates that effectiveness can outweigh efficiency in complex reasoning tasks such as coding, mathematics, and scientific discovery. The model achieves remarkable performance through requiring higher computational requirements during inference.
> >
> > Our ARMAP framework is an initial exploration of how such test-time scaling principles can extend to LLM agent domains. While it does incur additional computational overhead due to planning algorithms like MCTS, the ability to reason about and optimize decision trajectories aligns with the observed benefits of scaling in achieving better generalization and task performance.
> >
> > [1]. https://openai.com/o1/
> >
> > [2]. https://openai.com/index/learning-to-reason-with-llms/

---

> > > ### Author Response · Authors · 2024-11-25
> > > **Look forward for your further feedback (Part 3)**
> > >
> > > **Q3. Synthetic Data Quality.**
> > > > How do you ensure the quality and diversity of the synthetic trajectories generated by LLMs? Have you observed any limitations when these synthetic trajectories don’t align closely with real-world decision-making patterns?
> > >
> > > **Quality.** We use a capable LLM (LLaMA-70B) to generate data, and for different tasks, we have crafted distinct prompts that thoroughly describe the relevant environments and provide clear and precise instructions. Based on this approach, we can ensure that the quality of the generated data is not compromised. However, in rebuttal supplementary experiments, we have also demonstrated that our method can achieve good results even in low-resource scenarios using a smaller language model, Phi-3.8B.
> > >
> > > ||Seen|Unseen|
> > > |--------|--------|--------|
> > > |Greedy|29.9%| 23.8%
> > > |Phi-3.8B | 34.7%|26.9%
> > >
> > > Table 2-1: The experiments of sciworld on training data generated from smaller LLM.
> > >
> > > In the table above, even when we opt for the smaller Phi-3.8B model, we still achieve good experimental results. This demonstrates the quality of data generated using our method and shows that our approach does not require significant resources for data generation.
> > >
> > > **Diversity.** During the data generation process, we sometimes use methods like rephrasing to expand the diversity of instructions, thereby sampling more varied trajectories. Additionally, in the process of trajectory generation, we occasionally incorporate random walks to ensure that the trajectory content is not limited to certain parts but can explore different locations within the environment, thus ensuring trajectory diversity.
> > >
> > > **Observed Limitaion of Synthetic Trajectories.** At times, I observe examples with limitations that cannot complete the initial language instructions. In complex environments like ScienceWorld, the language agent requires robust scientific knowledge to complete scientific tasks within the environment. However, the steps of these scientific tasks are often lengthy, and each step requires strong scientific reasoning. Sometimes, the language agent may have hallucinations, leading to issues with the generated trajectory; other times, the language agent may terminate the trajectory prematurely, resulting in incomplete tasks. Nonetheless, with the ongoing development of large language models, these issues can be greatly alleviated in future work. It is worth noting that although we can identify some deficiencies during the data generation process, the overall quality of our data is quite good. We have implemented an efficient trajectory synthesis strategy through this automatic data generation method.
> > >
> > > **Q4. Computational Cost in Real-Time Applications**
> > > > Given the computational demands of planning algorithms like MCTS, how would ARMAP perform in applications requiring real-time decision-making? Are there strategies for reducing overhead while retaining performance?
> > >
> > > While our ARMAP, being grounded in planning algorithms like MCTS, is inherently designed for tasks requiring thorough exploration of decision pathways, making it highly effective in complex, high-stakes scenarios like science exporation, webshoping, robotic navigation, mathmetical problem-solving. However, we acknowledge that MCTS can introduce computational overhead that might hinder real-time performance. There might be some possible strategies for reducing overhead while retaining performance.
> > >
> > > **Parallelization and Hardware Acceleration.** Advances in parallel computing and specialized hardware (e.g., GPUs, TPUs) enable multiple simulations to run concurrently. Implementing a parallelized version of MCTS [1,2] within ARMAP can dramatically reduce decision-making latency without compromising performance. Furthermore, low-level optimizations for specific architectures could enhance processing speed.
> > >
> > > **Dynamic Planning Horizons.** ARMAP can adapt its planning depth or horizon dynamically based on the time constraints of the task. In time-critical scenarios, ARMAP could execute fewer simulations per decision step, sacrificing some precision for timeliness. For situations where more time is available, it can revert to deeper exploration.
> > >
> > > Note that we regard such improvement to make ARMAP for real-time applications as future work, since the primary goal of ARMAP's goal is to improve LLM's performance on scenarios that effectiveness is more important than efficiency like science exporation, webshoping, robotic navigation, mathmetical problem-solving.
> > >
> > > [1]. Chaslot, Guillaume MJ -B., Mark HM Winands, and H. Jaap van Den Herik. "Parallel monte-carlo tree search." Computers and Games: 6th International Conference. 2008.
> > >
> > > [2]. Steinmetz, Erik, and Maria Gini. "More trees or larger trees: Parallelizing Monte Carlo tree search." IEEE Transactions on Games.

---

> > > > ### Author Response · Authors · 2024-11-25
> > > > **Look forward for your further feedback (Part 4)**
> > > >
> > > > **Q5. Reward Model Generalization**
> > > > > How well does the reward model generalize to tasks and environments different from those it was trained on? Have you tested ARMAP in domains requiring more complex, domain-specific knowledge, such as legal or medical contexts?
> > > >
> > > > **Performance Adaption in New Environments.** Our ARMAP does show some capability to adapt to new patterns not present in the training data. This can be verified that our model is effective in both **seen** and **unseen** environments of ScienceWorld and ALFWorld. Note that the **unseen** split in these environments contain new patterns that has not been investigated in the training set. Our ARMAP framework has shown its effectiveness in Web Shoping (Webshop), Sciene Discovery (ScienceWorld), mathmetical problem solving (GameOf24) and Robotic Navigation (ALFWorld).
> > > >
> > > > However, we acknowledge that while ARMAP can generalize to new tasks within similar domains, building a truly general reward model capable of handling all scenarios across diverse environments remains a significant challenge. Addressing this is beyond the scope of this paper, but we regard it as a promising direction for future research.
> > > >
> > > > **Extension to New Domains.** To evaluate ARMAP in more complex, domain-specific settings, we conducted additional experiments during the rebuttal phase, focusing on ClinicalAgent [1], a benchmark environment designed for medical decision-making tasks. ClinicalAgent requires models to interpret clinical scenarios, reason accurately, and make high-stakes decisions in a domain where precision is critical. Results (Table 2-2) demonstrate that ARMAP adapts effectively to this domain, further supporting its versatility in environments requiring specialized knowledge and reasoning.
> > > >
> > > > | |AgentClinic-MedQA|
> > > > |--------|--------
> > > > |Sampling|11.89%|
> > > > |Greedy|14.02%|
> > > > |ARMAP-B |44.33%|
> > > >
> > > > Table 2-2: New experiments on AgentClinic.
> > > >
> > > > **Q6. Scalability and Practical Deployment.**
> > > > > What are the main challenges you foresee in scaling ARMAP for broader deployment in real-world applications? Are there specific areas (e.g., hardware requirements, integration with other models) that need further development?
> > > >
> > > > We appreciate the reviewer’s thoughtful question about the challenges associated with scaling ARMAP for broader deployment and its integration into real-world applications. While ARMAP has shown its value in complex scenarios where effectiveness outweighs efficiency, such as scientific reasoning, mathematics, and robotic navigation, we recognize several areas that require attention for practical deployment. Below, we outline key challenges and potential solutions.
> > > >
> > > > **Computational Demands.** ARMAP’s reliance on planning algorithms like MCTS and its need for iterative simulations can pose challenges in resource-constrained environments. High-performance hardware, such as GPUs or TPUs, are required to meet the computational demands for tasks with complex or large-scale decision spaces.
> > > > The rapid development of advanced computing units, such as GPUs and specialized accelerators (e.g., TPUs, FPGAs), combined with the ongoing decrease in hardware costs, is likely to make ARMAP’s computational requirements more accessible.
> > > >
> > > > **Adaptability to Diverse Real-World Scenarios.** Real-world applications often involve diverse, unstructured, or incomplete data inputs and environments with high variability. ARMAP’s performance depends on both the base LLM agent and the learned reward model. Current LLMs face challenges in handling such variability while ensuring robust generalization and adaptability. However, the continuous development of more powerful, multi-modal LLMs will enhance ARMAP’s ability to interpret and reason across diverse data types, including textual, visual, and structured data. The creation and collection of large-scale, domain-specific decision-making datasets will improve the reward model’s generalization capabilities.

---

> > > > > ### Author Response · Authors · 2024-11-28
> > > > > **Official Comment by Authors**
> > > > >
> > > > > Dear Reviewer **#XSp2**,
> > > > >
> > > > > Thank you for your valuable feedback, which has greatly contributed to improving our paper.
> > > > >
> > > > > We have addressed the reviewers’ comments in the author responses, submitted on November 25, 2024, and uploaded the latest version of our paper. We kindly invite you to review our detailed responses and let us know if there are any remaining concerns we can address before the discussion phase concludes.
> > > > >
> > > > > Regards,
> > > > >
> > > > > Authors of Submission 1719

---

> > > > > > ### Author Response · Authors · 2024-12-02
> > > > > > **Official Comment by Authors**
> > > > > >
> > > > > > Dear Reviewer **#XSp2**,
> > > > > >
> > > > > > Thank you for your constructive comments on our manuscript. In response to your suggestions, we have added further experiments and analyses to our paper, which are included in the updated version.
> > > > > >
> > > > > > With the rebuttal period ending in less than two days, we hope our revisions meet your expectations. If you are satisfied with our amendments, we would appreciate it if you could consider revising your score as other reviewers.
> > > > > >
> > > > > > We are grateful for your insightful feedback.
> > > > > >
> > > > > > Warm regards,
> > > > > >
> > > > > > Authors

---

### Official Review · Reviewer_G3nc · 2024-11-08

**Soundness:** 3
**Presentation:** 3
**Contribution:** 3
**Rating:** 6
**Confidence:** 3

**Summary:**

The paper proposes ARMAP, a novel framework that enhances the task-solving abilities of large language model (LLM)-based agents in interactive, multi-step environments. The authors tackle key challenges associated with data scarcity and API restrictions, presenting a method that automates reward model learning from LLM agents’ interactions within an environment, thus eliminating the need for human annotations or commercial LLM-based evaluation. The reward model can then guide planning algorithms (e.g., Monte Carlo Tree Search and Reflexion) to improve LLM agents’ performance in tasks requiring iterative decision-making, such as e-commerce navigation and simple scientific experiments.

**Strengths:**

Innovative Reward Modeling Approach: The ARMAP framework leverages LLMs to generate diverse action trajectories, then synthesizes task goals and feedback to train a reward model. This automation of reward modeling is a strong innovation, addressing critical limitations in agent-based tasks by reducing reliance on costly and often proprietary data.

Framework Flexibility: The framework’s compatibility with multiple planning algorithms (MCTS, Reflexion, Best-of-N) demonstrates flexibility and potential for broader application. The performance boost across different LLMs (Llama, Phi, and Mistral) also underscores the generalizability of the ARMAP model.

Effectiveness in Customization: ARMAP’s ability to modify reward targets for controllable behavior generation (e.g., minimizing action length or cost) is a valuable capability for task-specific tuning, as demonstrated in the Webshop experiments.

**Weaknesses:**

Limited Scope of Tested Environments: Although the ARMAP framework was evaluated in multiple environments, these remain relatively constrained in task diversity (e.g., online shopping, elementary science tasks). Further exploration into environments with more complex multi-modal interactions or requiring intricate goal alignment would provide stronger evidence of the framework’s versatility.

Potential Overhead in Data Synthesis: While the automated reward modeling is valuable, the reliance on in-context LLMs for both task generation and trajectory synthesis could introduce computational overhead. It would be useful to discuss the cost-benefit analysis of this approach, particularly in environments requiring higher levels of interaction fidelity.

Dependence on LLM Quality: ARMAP’s effectiveness is inherently tied to the quality of the LLMs generating the synthetic data. While the framework was evaluated on open-source models, a more explicit discussion of performance across varying LLM qualities or limitations when using smaller LLMs would provide more insight into its applicability in resource-constrained scenarios.

**Questions:**

Some suggestions for improvement:

Why do we need pairwise comparisons - this works in foundation model post-training, but why not use success/failure reward model training and using that as areward or value function?

Can you extend the experimental scope to include more diverse or high-stakes decision-making environments, such as ALFRED, BEHAVIOUR or HABITAT to illustrate ARMAP’s performance on tasks requiring more advanced capability.

Computational Efficiency Analysis: Including an analysis of the framework's data demands and comparisons with reward learning approaches would be beneficial, especially if extending the applicability of ARMAP to realistic low-resource settings.

Detailed Error Analysis: A more granular analysis of failure cases in each environment, particularly for tasks that involve complex dependencies or decision making, would provide deeper insights into the limitations of the current approach and inform possible improvements in reward modeling.

---

> ### Author Response · Authors · 2024-11-25
> **Look forward for your further response (Part 1)**
>
> **Q1. Experiments in More Environments.**
> >Although the ARMAP framework was evaluated in multiple environments, these remain relatively constrained in task diversity (e.g., online shopping, elementary science tasks). Further exploration into environments with more complex multi-modal interactions or requiring intricate goal alignment would provide stronger evidence of the framework’s versatility.
> >Can you extend the experimental scope to include more diverse or high-stakes decision-making environments, such as ALFRED, BEHAVIOUR or HABITAT to illustrate ARMAP’s performance on tasks requiring more advanced capability.
>
> We thank the reviewers for suggestions on more experiments in new and more diverse environments. During rebuttal, we have added new experiments in ALFWorld[1] and ClinicalAgent[2].
>
> **Effectiveness of Our Pipeline.** We provide detailed explanation about the effectiveness of our pipeline in **G2**. There, we extend our method into two different tasks in two different domains. These new experimental results confirm the generalizability, effectiveness, and adaptability of ARMAP.
>
> **Future Work on Visual-Dominated Environments.** While our current experiments focus on environments with textual or limited visual contexts, such as webshops (e.g., shopping assistants), ScienceWorld, and GameOf24, we recognize the importance of evaluating ARMAP in more visually dominated environments, such as BEHAVIOUR and Habitat. These environments involve complex visual understanding and multi-modal interactions, posing unique challenges for ARMAP’s current framework.
> At present, our pipeline relies on open-source models such as LLaMA-3 and Mixtral, which primarily support textual inputs. While these models perform effectively in textual-centric environments, their limitations prevent us from fully leveraging ARMAP in heavily visual environments. However, we believe that as open-source foundation models evolve to support multi-modality, ARMAP can be extended to tackle these tasks. Future multi-modal models will enable us to generate richer training data and better integrate visual understanding into ARMAP, allowing it to scale effectively to environments like Habitat and BEHAVIOUR.
>
> **Value of Current Scope.** Even within textual and limited-visual environments, ARMAP demonstrates significant utility for a variety of applications, including shopping assistants, scientific reasoning, mathematical problem solving, and robotic navigation. These environments serve as a strong foundation for further development, showcasing the robustness and adaptability of ARMAP's planning-based framework in diverse scenarios.
>
> [1] Mohit Shridhar, et al. Alfworld: Aligning text and embodied environments for interactive learning. In International Conference on Learning Representations, 2020.
>
> [2] S. Schmidgall, R. Ziaei, C. Harris, E. Reis, J. Jopling, and M. Moor, “Agentclinic: a multimodal agent benchmark to evaluate AI in simulated clinical environments,” 2024.

---

> > ### Author Response · Authors · 2024-11-25
> > **Look forward for your further response (Part 2)**
> >
> > **Q2. Overhead in Data Synthesis.**
> > >While the automated reward modeling is valuable, the reliance on in-context LLMs for both task generation and trajectory synthesis could introduce computational overhead. It would be useful to discuss the cost-benefit analysis of this approach, particularly in environments requiring higher levels of interaction fidelity.
> >
> > Thanks for the suggestion to discuss the cost-benefit analysis of our approach. During the rebuttal, we have calculated the tokens we have used for task instruction generation and trajectory exploration. We summarize these overhead in Table 1-1.
> >
> > In the Table 1-1, we calculated the number of samples and tokens used in different tasks. In order to have a more direct comparison, we compare our method with the consumption of GPT4 and Amazon Mechanical Turk. The results support that our proposed method is efficient and cost-effective.
> >
> > |  Tasks   | Samples | Tokens   | Tokens per Sample |
> > | -------- | ------- | -------- | ----------------- |
> > | ScienceWorld | 4064    | 2541255  |        625        |
> > | Webshop  | 2436    | 6645746  |        2728       |
> > | GameOf24   | 37885   | 12846182 |        339        |
> >
> > Table 1-1: Tokens of data generation in three different tasks
> >
> > To provide a more intuitive comparison, we first calculated the average tokens per sample for these different tasks. We found that although GameOf24 overall consumes the most tokens, the average number of tokens spent per GameOf24 sample is relatively the least. In contrast, Webshop has the fewest total samples but the highest average number of tokens spent per sample. ScienceWorld falls in between these two. The reason webshop has a higher average number of tokens compared to GameOf24 is that the environment required for Webshop is more complex, involving more diverse elements and possibilities.
> >
> > Next, we can make a rough comparison with GPT-4 and Amazon Mechanical Turk (AMT) to demonstrate our advantages.
> >
> > (1) GPT4: The input and output token prices for GPT4 are 30 dollars per 1M tokens and 60 dollars per 1M tokens, respectively. Taking an average, that would be 45 dollars per 1M tokens. With this rate, we would need to spend 114, 299, and 578 dollars to generate data for ScienceWorld, Webshop, and GameOf24, respectively. These prices are quite high for experimental data generation. Additionally, we use open-source models that, apart from being free and not requiring additional manual annotations, is also more transparent, allowing for better analysis of the data generation results.
> >
> > (2) AMT: It is difficult to generalize the pricing for AMT, but using human labor often incurs higher costs. Moreover, if we use AMT, our tasks would require complex annotations. This means that annotators would need more time for annotation, and we would need to spend a lot of money. Often, despite these costs, the data quality collected via AMT can be poor.

---

> > > ### Author Response · Authors · 2024-11-25
> > > **Look forward for your further response (Part 3)**
> > >
> > > **Q3. Dependence on LLM Quality.**
> > > >ARMAP’s effectiveness is inherently tied to the quality of the LLMs generating the synthetic data. While the framework was evaluated on open-source models, a more explicit discussion of performance across varying LLM qualities or limitations when using smaller LLMs would provide more insight into its applicability in resource-constrained scenarios.
> > >
> > > During rebuttal, we choose ScienceWorld and have conducted new experiments to study the effectiveness of different reward models. As shown in Table 1-2, the left column represents the results of using LLaMA-8B greedy directly and the Best of N results of LLaMA-8B with the reward model trained by the data generated from LLaMA-70B, LLaMA-8B, Mistral-7B, and Phi-3.8B, respectively. We found that owing to our automatic data generation pipeline, the data generated from different sizes of LLMs can be well used to train effective reward models.
> > >
> > > ||Seen|Unseen|
> > > |--------|--------|--------|
> > > |Greedy|29.9%| 23.8%
> > > |LLaMA-70B|35.7%|28.1%
> > > |LLaMA-8B|32.2%|24.7%
> > > |Mistral-7B|33.7%|26.5%
> > > |Phi-3.8B | 34.7%|26.9%
> > >
> > > Table 1-2: New experiments on training data generated from various LLMs.
> > >
> > > In the table above, Greedy is our baseline result. It can be observed that using the reward model leads to better experimental outcomes. Among all the results, LLaMA-70B achieves the best performance. Compared to the other three models, LLaMA-70B has the largest scale and is naturally the most capable model. LLaMA-8B and Mistral-7B have a similar number of parameters, and in the ScienceWorld task, Mistral-7B performs better than LLaMA-8B. Phi-3.8B is the smallest of these models, yet it still achieved very good results. Notably, compared to the larger-scale LLaMA-8B and Mistral-7B, Phi-3.8B still scored better. These results indicate that our method exhibits good robustness when faced with LLMs of different scales and capabilities. Even with the smallest model, our method can still achieve good results. From these experimental outcomes, it is clear that our method does not overly rely on the capabilities of language models. In other words, our method is highly efficient and robust.

---

> > > > ### Author Response · Authors · 2024-11-25
> > > > **Look forward for your further response (Part 4)**
> > > >
> > > > **Q4. Choices for Reward Modeling Target.**
> > > > > Why do we need pairwise comparisons - this works in foundation model post-training, but why not use success/failure reward model training and using that as a reward or value function?
> > > >
> > > > Thank you for raising this important question about the optimization target of the reward model. We chose pairwise comparison as the reward target for the following reasons.
> > > >
> > > > **Capturing Relative Preferences.** Pairwise comparison focuses on learning relative preferences between two trajectories, rather than classifying each trajectory (success/failure) independently. This approach is particularly effective in scenarios where the goal is to identify which of two responses is preferred, as it captures nuanced differences between trajectories. For instance, it can easily distinguish that a trajectory with a reward value of 0.6 is better than another with a reward of 0.5. In contrast, labeling such trajectories as "success" or "failure" in a binary classification scheme may lead to oversimplification and loss of information about the subtle gradation in quality.
> > > >
> > > > **The Bradley-Terry Model.** In our implementation, we use the Bradley-Terry model [1], a widely adopted method for pairwise comparisons. This model estimates the probability of one trajectory being preferred over another, making it especially suitable for ranking tasks. By directly modeling preferences, the Bradley-Terry framework enables us to train the reward model to capture more nuanced differences in trajectory quality.
> > > >
> > > > **Alignment with State-of-the-Art Practices.** Pairwise comparison is a common optimization target in recent popular LLM models [2, 3, 4, 5], further validating its effectiveness. These models have demonstrated that relative preference learning outperforms classification in tasks requiring fine-grained judgment.
> > > >
> > > > **Additional Experiments.** To further investigate this, we conducted new experiments during the rebuttal phase to compare the performance of pairwise comparison and binary classification as learning methods for the reward model. Specifically, in the classification setting: Each input pair is treated as a positive and a negative example. The model is trained to predict a score of 1 for positive examples and 0 for negative examples.
> > > > The comparative results are shown in Tables 1-3 to 1-6. Across all settings, pairwise comparison consistently outperforms binary classification. This confirms that pairwise comparison captures nuanced preferences more effectively than binary classification, leading to better reward modeling and overall task performance.
> > > >
> > > > |ARMAP-B |Classification|Comparative|
> > > > |--------|--------|--------|
> > > > |LLaMA-70B|47.2%|57.3%
> > > > |LLaMA-8B|27.5%|35.7%
> > > > |Mistral-7B|19.1%|24.5%
> > > > |Phi-3.8B | 17.7%|20.0%
> > > >
> > > > Table 1-3: Comparison of the Clssification target and Comparison target on ScienceWorld Seen Split with ARMAP-B.
> > > >
> > > > |ARMAP-R |Classification|Comparative|
> > > > |--------|--------|--------|
> > > > |LLaMA-70B|57.0%|59.0%
> > > > |LLaMA-8B|29.0%|31.2%
> > > > |Mistral-7B|17.8%|21.7%
> > > > |Phi-3.8B | 8.6%|9.6%
> > > >
> > > > Table 1-4: Comparison of the Clssification target and Comparison target on ScienceWorld Seen Split with ARMAP-R.
> > > >
> > > > |ARMAP-R |Classification|Comparative|
> > > > |--------|--------|--------|
> > > > |LLaMA-70B|43.3%|57.0%
> > > > |LLaMA-8B|22.2%|28.1%
> > > > |Mistral-7B|17.3%|21.1%
> > > > |Phi-3.8B | 13.7%|17.0%
> > > >
> > > > Table 1-5: Comparison of the Classification target and Comparison target on ScienceWorld Unseen Split with ARMAP-B.
> > > >
> > > >
> > > > |ARMAP-R |Classification|Comparative|
> > > > |--------|--------|--------|
> > > > |LLaMA-70B|55.4%|56.7%
> > > > |LLaMA-8B|24.2%|28.0%
> > > > |Mistral-7B|18.2%|19.7%
> > > > |Phi-3.8B | 4.8%|7.2%
> > > >
> > > > Table 1-6: Comparison of the Clssification target and Comparison target on ScienceWorld Unseen Split with ARMAP-R.
> > > >
> > > > [1]. Bradley, Ralph Allan, and Milton E. Terry. "Rank analysis of incomplete block designs: I. The method of paired comparisons." Biometrika 39.3/4 (1952): 324-345.
> > > >
> > > > [2]. Touvron, Hugo, et al. "Llama 2: Open foundation and fine-tuned chat models." arXiv preprint arXiv:2307.09288 (2023).
> > > >
> > > > [3]. Dubey, Abhimanyu, et al. "The llama 3 herd of models." arXiv preprint arXiv:2407.21783 (2024).
> > > >
> > > > [4]. Yang, An, et al. "Qwen2 technical report." arXiv preprint arXiv:2407.10671 (2024).
> > > >
> > > > [5]. Sun, Zhiqing, et al. "Principle-driven self-alignment of language models from scratch with minimal human supervision." Advances in Neural Information Processing Systems 36 (2024).

---

> > > > > ### Author Response · Authors · 2024-11-25
> > > > > **Look forward for your further response (Part 5)**
> > > > >
> > > > > **Q5. Computational Efficiency Analysis.**
> > > > > >Including an analysis of the framework's data demands and comparisons with reward learning approaches would be beneficial, especially if extending the applicability of ARMAP to realistic low-resource settings.
> > > > >
> > > > > Thanks for the suggestion on more analysis of data demands and comparisons with different reward modeling approaches. In previous response to **Q4**, we have provided detailed experiments for different approaches to model rewards. In this question, we further study the data demands of the reward modelings. We show the performance of using different amount of training data in Table 1-7 and Table 1-8.
> > > > >
> > > > > | ARMAP-B  | VILA-3B    | VILA-13B   | LLaVA-13B | 1/5 Data | 1/25 Data |
> > > > > | -------- | ----- | ----- | ----- | -------- | --------- |
> > > > > | LLaMA-70B | 57.3% | 61.2% | 44.3% | 52.1%    | 50.6%     |
> > > > > | LLaMA-8B  | 35.7% | 34.3% | 26.0% | 31.4%    | 29.3%     |
> > > > > | Mistral-7B  | 24.5% | 26.0% | 19.5% | 22.6%    | 21.7%     |
> > > > > | Phi-3.8B      | 20.0% | 19.5% | 16.7% | 17.9%    | 13.9%     |
> > > > >
> > > > > Table 1-7: The effects of our ARMAP-B method under different policy models in the context of varying reward models and training data volumes in ScienceWorld Seen.
> > > > >
> > > > > | ARMAP-B  | VILA-3B    | VILA-13B   | LLaVA-13B | 1/5 Data | 1/25 Data |
> > > > > | -------- | ----- | ----- | ----- | -------- | --------- |
> > > > > | LLaMA-70B | 57.0% | 60.7% | 48.2% | 50.0%    | 47.7%     |
> > > > > | LLaMA-8B  | 28.1% | 27.5% | 22.2% | 26.8%    | 24.2%     |
> > > > > | Mistral-7B  | 21.1% | 22.9% | 19.2% | 21.6%    | 19.7%     |
> > > > > | Phi-3.8B      | 17.0% | 15.3% | 13.7% | 14.2%    | 11.7%     |
> > > > >
> > > > > Table 1-8: The effects of our ARMAP-B method under different policy models in the context of varying reward models and training data volumes in ScienceWorld Unseen.
> > > > >
> > > > > In the table above, we selected ScienceWorld and used ARMAP-B as the experimental setting. In the leftmost column, we listed the different LLMs used in our study. In the first row, we introduced VILA-3B, VILA-13B, and LLaVA-13B, to compare the impact of different sizes and types of reward models on the final outcomes. In the last two columns, we trained the reward models using 1/5 and 1/25 of the original training dataset size, respectively, to assess how varying amounts of training data affect our method. (1) As seen, the effectiveness of our method continues to improve with increasing reward model sizes. However, in the experiments with LLaMA-8B and Phi-3.8B, despite using more potent reward models, there was no improvement in results. We believe that in the processes of planning and reasoning, the capability of the policy model still plays a dominant role. If the policy model is more robust, and concurrently, if we enhance the capability of the reward model, we can continuously achieve better results. (2) We also observe that the performance of LLaVA-13B is not as good as VILA-13B. We attribute this to VILA being an improved version of LLaVA, and it utilizes an interleaved image-text dataset in its training, which better aids the model in perceiving, understanding, and handling multimodal information. Hence, VILA outperforms LLaVA. (3) From the table, it is evident that regardless of whether the data is seen or unseen, increasing the model size improves the final experimental results. If we use the results of the VILA-3B model as a benchmark and compare it with the two settings, 1/5 data and 1/25 data, it is clear that increasing the training data enhances the outcomes. Conversely, even when using extremely limited data amounts like 1/5 or 1/25 of the original dataset, we can still achieve a capable model, and the performance does not dramatically decrease.
> > > > >
> > > > > These results demonstrate that our method can still yield good results in a low-resource environment. In other words, our approach does not rely on large volumes of data and the strong capability of large models; it is succinct and efficient, capable of performing well in extremely low-resource settings.
> > > > >
> > > > > **Q6. Detailed Error Analysis.**
> > > > > >A more granular analysis of failure cases in each environment, particularly for tasks that involve complex dependencies or decision making, would provide deeper insights into the limitations of the current approach and inform possible improvements in reward modeling.
> > > > >
> > > > > Thanks for the reminder to provide analysis of failure cases in each environment. During rebuttal, we have added subsection in the revised paper to study the failure cases of the model. To be brief, we find that if the instruction contains many complex and detailed conditions or the instruction requires ample commonsense knowledge, the agent is more likely to make mistakes. For more details, please refer to failure cases section of the revised paper.

---

> > > > > > ### Author Response · Authors · 2024-11-28
> > > > > > **Official Comment by Authors**
> > > > > >
> > > > > > Dear Reviewer **#G3nc**,
> > > > > >
> > > > > > We would like to thank you for your helpful feedback which has helped us improve the paper.
> > > > > >
> > > > > > We addressed reviewers' concerns in the author's responses, posted on the 25th of Nov 2024. We also uploaded the latest version of our paper. We would be delighted if you could please take a look at our detailed responses so that we can address any remaining concerns before the end of the discussion phase.
> > > > > >
> > > > > > Sincerely,
> > > > > >
> > > > > > Authors of Submission 1719

---

> > > > > > > ### Author Response · Authors · 2024-12-02
> > > > > > > **Official Comment by Authors**
> > > > > > >
> > > > > > > Dear Reviewer **#G3nc**,
> > > > > > >
> > > > > > > We sincerely thank you for your valuable feedback on our manuscript. Following your suggestions, we have enriched our paper with additional experiments and analysis, which are now included in the revised manuscript.
> > > > > > >
> > > > > > > As the rebuttal period concludes in less than two days, we hope our efforts align with your expectations. If you find our response satisfactory, we would be grateful if you could consider revising your score as other reviewers.
> > > > > > >
> > > > > > > Thank you once again for your insightful guidance.
> > > > > > >
> > > > > > > Warm regards,
> > > > > > >
> > > > > > > Authors

---

### Author Response · Authors · 2024-11-25
**General Response: Contributions, Novelty, Broader Task Scope, New Experiments and Paper Revision (Part 1)**

**G1. Contribution Recognition.**

We extend our sincere gratitude to the reviewers for their time and effort in reviewing our paper. We are pleased to note that the reviewers have generally acknowledged the ARMAP's following contributions:

* **the automatic reward modeling approach is novel.** This automation of reward modeling is a strong innovation, addressing critical limitations in agent-based tasks by reducing reliance on costly and often proprietary data **(G3nc)**; it presents an innovative method for autonomously learning reward models without the need for human-annotated data, addressing issues related to data scarcity and dependence on costly closed-source LLMs **(XSp2)**; the automatic reward model and data generation approach presented is novel, allowing the framework to guide task completion within complex decision-making environments effectively **(Rvsi)**.

* **the propsoed ARMAP framework is effective and efficient.** By offering a reward-based evaluation system, ARMAP significantly boosts the ability of LLM agents to perform complex, multi-step tasks that require sequential planning, an area where standard LLMs often struggle **(XSp2)**; by eliminating the need to fine-tune LLMs and avoiding reliance on proprietary LLM APIs, ARMAP provides a cost-effective solution that could make high-performing AI agents more accessible for widespread use **(XSp2)**; ARMAP stands out by using a reward model to evaluate and guide navigation steps in agentic environments, enhancing decision-making processes and setting a solid foundation for handling intricate tasks autonomously **(Rvsi)**.

* **the propsoed ARMAP framework is flexible and customizable.** The framework’s compatibility with multiple planning algorithms (MCTS, Reflexion, Best-of-N) demonstrates flexibility and potential for broader application. The performance boost across different LLMs (Llama, Phi, and Mistral) also underscores the generalizability of the ARMAP model **(G3nc)**; ARMAP’s ability to modify reward targets for controllable behavior generation (e.g., minimizing action length or cost) is a valuable capability for task-specific tuning, as demonstrated in the Webshop experiments **(G3nc)**; the framework's value is demonstrated through LLM-agent task performance, highlighting flexibility in controllable task generation and practical application via a reward model, which reduces reliance on large LLMs or human labeling **(Rvsi)**.

---

> ### Author Response · Authors · 2024-11-25
> **General Response: Contributions, Novelty, Broader Task Scope, New Experiments and Paper Revision (Part 2)**
>
> **G2. Our Task Scope.**
>
> Thanks for the constructive comments from the reviewers. During the rebuttal period, we added more tasks to verify that our method can be effective in a broader and more complex range of tasks.
>
> We extend our experiment on ALFWorld [1], a classic environment for House-Holding, where the agent must accomplish tasks in physical house-holding environments, like “Put a pan on the dining table”. Following the setup of AgentBench [2] for LLM evaluation, we test the model on the Dev and Std split, using the default success rate as the evaluation metric.
>
> Specifically, we used LLaMA-3.1-70B to generate around 1600 pairs of positive and negative samples with our data generation pipeline. Then we train a reward model with these synthesized data. We evaluate our ARMAP framework on ALFWorld using various planning algorithms, including Reflexion and Best-of-N, which we refer to as ARMAP-R and ARMAP-B, respectively. Additionally, we compare our approach with two baseline methods that do not incorporate reward model guidance: Sampling and Greedy. The results are shown below. As shown in Table G-1, our model still performs well in this challenging environment, which contains diverse scenes and long-horizon planning tasks.
>
> We also extended our experiments to ClinicalAgent [3], an environment designed for medical decision-making tasks. ClinicalAgent evaluates models on their ability to interpret clinical scenarios and make accurate, high-stakes decisions. Results for ClinicalAgent are provided in table G-2, further supporting the versatility of ARMAP in domains requiring precise reasoning.
>
> | |Std|Dev
> |--------|--------|--------|
> |Sampling|0.13|0.14
> |Greedy|0.18|0.30
> |ARMAP-R|0.22|0.35
> |ARMAP-B | 0.30|0.45
>
> Table G-1: New experiments on ALFWorld.
>
> | |AgentClinic-MedQA|
> |--------|--------
> |Sampling|11.89%|
> |Greedy|14.02%|
> |ARMAP-B |44.33%|
>
> Table G-2: New experiments on AgentClinic.
>
> ARMAP demonstrates significant utility for a variety of applications, including shopping assistants, scientific reasoning, mathematical problem solving, embodied agentic reasoning and planning, and medical decision-making. These diverse environments serve as a strong foundation for further development, showcasing the robustness, generalizability, and adaptability of ARMAP's planning-based framework in different scenarios.
>
> In these extensive tasks, our proposed method also achieves good results. This demonstrates that our method is not only applicable to the limited tasks discussed in the paper but can also be easily transferred to more diverse tasks across different domains.
>
> [1] Mohit Shridhar, et al. Alfworld: Aligning text and embodied environments for interactive learning. In International Conference on Learning Representations, 2020.
>
> [2] Xiao Liu, et al. Agentbench: Evaluating llms as agents. arXiv preprint arXiv: 2308.03688, 2023.
>
> [3] S. Schmidgall, R. Ziaei, C. Harris, E. Reis, J. Jopling, and M. Moor, “Agentclinic: a multimodal agent benchmark to evaluate AI in simulated clinical environments,” 2024.
>
> **G3. Experiments in the revision.**
>
> To address the reviewers’ concerns and support our responses, we conduct the following experiments to support our claims:
> * more tasks exploration: (1) experiments on ALFWorld, (2) experiments on ClinicalAgent; **(G3nc, XSp2, Rvsi)**
> * analysis of efficiency of ARMAP: (3) overhead in data synthesis, (4) dependence on LLM quality, (5) experiments of data demands of our approach; **(G3nc, XSp2)**
> * more details of reward modeling: (6) choices for reward modeling target, (7) experiments of different sizes and types of reward models; **(Rvsi)**
> * (8) ablation on visual input. **(Rvsi)**
>
> **G4. Paper Revision.**
>
> Besides the experimental results, we have also revised the paper correspondingly (highlighted in blue);
> * include experimental results into the revised version of the paper;
> * revise typos in the paper.

---

### Meta-Review · Area_Chair_ntEb · 2024-12-19

**Metareview:**

The paper proposes learning a reward model using an LLM-based agent's multi-step trajectories, and using the learned reward model with planning (e.g. Monte Carlo tree search) to improve the performance of LLM-agents on multi-step tasks.

All of the reviewers agreed that the paper is above the bar for publication at ICLR. The approach was deemed to be effective, and addressing a well motivated problem.

The reviewers identified several missing details (e.g. token costs of data gathering, reward model architecture) and suggested several additional experiments. The authors provided these details and ran additional experiments during the rebuttal to adequately answer these questions -- these additional information should be included in a substantial revision of the paper to strengthen it.

**Additional Comments On Reviewer Discussion:**

Reviewers identified a few weaknesses:
1. Limited diversity in the domains for the empirical studies. To answer this, the authors ran additional experiments in AlfWorld and MedQA (AgentClinic).

2. Insufficient discussion on the overheads/costs of the proposal. The authors provided details on the tokens used to generate the data across the different domains.

3. Ablations on the reward modeling. The authors varied the model used to generate the trajectories data, and showed the resulting performance after training the reward model on the different datasets, as well as using different reward model architectures (VILA, LLAVA, etc.)

4. Variations on the training objective. The authors compared their proposal of training on contrastive trajectories against a reviewer's suggestion of success/failure classifier trained on the trajectories, and they concluded that the contrastive training was marginally better in their domains.

The authors' comprehensive rebuttal and the additional experimental results and details were instrumental in improving the quality of the submission above the bar for publication at ICLR.

---

### Decision · Program_Chairs · 2025-01-22

Accept (Poster)